# Ash Characteristics and Selected Critical Elements (Ga, Sc, V) in Coal and Ash in Polish Deposits

Barbara Bielowicz

Faculty of Geology, Geophysics and Environment Protection, AGH University of Science and Technology, Al. Mickiewicza 30, 30-059 Kraków, Poland; bbiel@agh.edu.pl

**Abstract:** The chemical composition of coal ash and the content of the critical elements Ga, Sc, and V in coal and ash are examined herein. In this study, lignite and bituminous coal from Polish deposits were used. The coals were subjected to ultimate and proximate analysis; the petrographic composition was determined based on maceral groups. The chemical composition of ash and the content of critical elements were determined using ICP-MS. The obtained results were correlated and Pearson's linear correlation coefficient was determined. Based on the correlation analysis, the relationship between the chemical composition of ash and the proximate and ultimate analyses was demonstrated. The content of selected critical elements in the tested deposits was lower than the Clarke value in coal. However, in some deposits these contents are much higher in coal ashes. The higher levels of Ga, V, and Sc in the ash are associated with $Al_2O_3$. Therefore, it can be stated that ashes can be a potential source of some raw materials. The highest concentrations of critical elements in coal and ash were recorded in the Lublin Coal Basin. Supra-Clarke contents of Ga, V, and Sc were recorded in the Bogdanka coal mine.

**Keywords:** coal; coal ash; critical elements; gallium; scandium; vanadium

## 1. Introduction

Poland, like China, Greece, and Turkey, is one of those countries where electricity generation is still based on coal. Both lignite and bituminous coal are mined in Poland. Therefore, the amount of waste generated from their combustion is relatively high. In Poland, a total of about 20 million tons of ash, also known as "by-products", are generated by lignite and bituminous coal power plants each year. These products can be utilized in many ways [1–4]. They are used for reclamation, as a backfilling material for post-mining voids, or in the construction of streets and roads. They are used in the cement and concrete industries, and as a raw material for the production of building materials (e.g., cellular concrete blocks and bricks). Ashes with a high calcium content are good fertilizers. They can be used for the production of paints and varnishes. However, there are ongoing efforts aimed at finding new possible uses for ashes. One of them is the recovery of critical raw materials in the ash. In 2008, the Committee on Critical Mineral Impacts on the U.S. Economy introduced the term "critical raw materials". The term includes raw materials for which supply interruption would have the most harmful consequences for the economy. According to the "Communication on the list of critical raw materials" [5], 20 critical raw materials were identified. In 2017, another assessment was performed for 78 raw materials. The assessment contained nine new raw materials compared to the 2014 assessment. According to the EU [6], critical raw materials are antimony, baryte, beryllium, bismuth, borate, cobalt, coking coal, fluorspar, gallium, germanium, hafnium, helium, indium, magnesium, natural graphite, natural rubber, niobium, phosphate rock, phosphorus, scandium, silicon metal, tantalum, tungsten, vanadium, platinum group metals, heavy rare earth elements, and light rare earth elements. While heavy rare-earth elements (HREY) (dysprosium, erbium, europium, gadolinium, holmium,

lutetium, terbium, thulium, ytterbium, yttrium), light rare-earth elements (LREY) (lanthanum, cerium, praseodymium, neodymium, and samarium), and metals from the platinum group (palladium, platinum, rhodium, ruthenium, iridium) were assessed individually, they were still compiled in groups in the criticality list. These 27 raw materials are critical to the EU, as risks of supply shortage and their effects on the economy are higher than those of most of the other raw materials. China is the largest supplier of most of critical raw materials, including rare earth elements, magnesium, tungsten, antimony, gallium, and germanium. The remaining major suppliers of certain raw materials include Brazil (niobium) or the USA (Be and He). The f metals from the platinum group are mostly supplied by Russia (Pd) and South Africa (Ir, Pt, Rh, and Ru). The risk associated with the concentration of sourcing is usually compounded by a low level of substitution and low recycling levels.

The second key issue is the recovery and extraction of metals and critical elements, which are widely used in modern technologies and their prices are still increasing. More attention should be paid to the ever-increasing demand for REE (rare-earth element). The total amount of REE global resources is 478 Mt REO (rare earth oxides), located in China (164 Mt), Brazil (55 Mt), Australia (49 Mt), Russia (48 Mt), and Greenland (43 Mt), with the remaining 119 Mt scattered throughout Canada, Sweden, USA, Vietnam, and others. Rare earth elements are necessary for the development of, among other things, wind energy, linear fluorescent lamps (LFLs), compact fluorescent lamps (CFLs), light-emitting diodes (LEDs), electric vehicles (EVs), electric bicycles, and NiMH batteries and catalysts. Therefore, recovering these resources from waste is one of the main principles of the so-called urban mining and circular economy. Taking into account numerous applications of critical elements and raw materials, an emphasis should be placed on the production of clean energy. The most promising elements include Ga, In, REE, Si, Pt, Li, Co, Ge, Ru, Pd, and Se [7].

Coal and its ashes, including the fly ash collected in electrostatic precipitators of power plants and CHPs (Combined Heat and Power), are a source of critical raw materials despite the fact that the coal and waste generated during its exploitation are not classified as critical raw materials. The reason for this is the lack of studies. This is due to the low concentration of critical raw materials in coal, which is similar to the average content in other sedimentary rocks (the Clarke value of sedimentary rocks). Economically viable concentrations of critical raw materials in the abovementioned raw materials are rare and are observed mainly in small deposits formed or subjected to secondary transformation under specific sedimentation conditions (e.g., epigenetic hydrothermal and/or exhalation mineralization or low metamorphism, especially during metasomatic metamorphism). The mineralogical and elemental compositions of coal combustion remains have been the subject of several studies during the last three decades due to technical and environmental concerns [8–13]. The possibility of obtaining critical elements was first discussed in the context of ashes from Russian coals [14,15]. The question of critical raw materials in coal and its ashes has been mainly analyzed by scientists from China. This is related to critical elements found in coal deposits [16]. The question of critical raw materials in coal has been discussed by Dai et al. [17,18], Lefticariu et al. [19], Lin et al. [20], Seredin and Dai [21], and Tang et al. [22]. The content of critical raw materials in coal ashes was analyzed by, among others, Bielowicz et al. [23], Dai et al. [24], Franus et al. [25], Kataka et al. [26], Kolker et al. [27], and Valentim et al. [28]. Based on the literature data on the content of critical elements in coal and its ashes, it can be unequivocally stated that further studies are required.

This paper focuses on the share and composition of selected elements in relation to various coal lithotypes. Until recently, the chemical composition was only tested for the whole coal. The reason for using samples of different lithotypes was the fact that the petrographic composition of coal, and, thus, its lithotype, is associated with the environment of its sedimentation. This environment also affects the composition of the mineral matter and the content of critical elements in coal and ash.

## 2. Materials and Methods

A total of 28 coal samples, including 13 low-rank coals from six deposits and 15 bituminous coal samples from five deposits, were analyzed. Lignite samples were collected from the Miocene deposits

of the Polish Lowland. The subbituminous coal sample was from the Jurassic age and was collected from the Poręba deposit in the Silesian–Kraków monocline. Bituminous coal from the Upper Silesian Coal Basin and Lublin Coal Basin was formed during the Upper Carbonaceous period (Table 1).

**Table 1.** Location, age, and lithotype of coal samples.

| Rank | Sample No. | Deposit |
|---|---|---|
| Lignite | 1L | Pątnów (Jóźwin opencast), Middle Miocene (1st seam), detritic |
| | 2L | Pątnów (Jóźwin opencast), Middle Miocene (1st seam), xylitic |
| | 3L | Pątnów (Jóźwin opencast), Middle Miocene (1st seam), detro-xylitic |
| | 4L | Turów, Early Miocene (3rd seam), xylitic |
| | 5L | Turów, Early Miocene (3rd seam), xylo-detritic |
| | 6L | Sieniawa, Early-Middle Miocene (2nd seam), xylo-detritic |
| | 7L | Sieniawa, Early-Middle Miocene (2nd seam), detritic |
| | 8L | Bełchatów, Early Miocene (3rd seam), xylo-detritic |
| | 9L | Bełchatów, Lower Miocene, xylitic |
| | 10L | Bełchatów, Early Miocene (3rd seam), detritic |
| | 11L | Szczerców, Early-Middle Miocene (2rd seam), xylitic |
| | 12L | Szczerców, Early-Middle Miocene (2rd seam), xylo-detritic |
| Subbituminous coal | 1SB | Poręba, Early Jurassic, detritic |
| Bituminous coal | 1B | Jas-Mos, Westphalian A (seam 510), coking coal |
| | 2B | Janina, Westphalian D (seam 119/2), bright coal |
| | 3B | Janina, Westphalian D (seam 119/2), dull coal |
| | 4B | Janina, Westphalian C (seam 203/2), bright coal |
| | 5B | Janina, Westphalian C (seam 203/2), dull coal |
| | 6B | Bogdanka, Westphalian B (seam 391), bright coal |
| | 7B | Bogdanka, Westphalian B (seam 391), dull coal |
| | 8B | Bogdanka, Westphalian B (seam 381), bright coal |
| | 9B | Bogdanka, Westphalian B (seam 381), dull coal |
| | 10B | Bogdanka, Westphalian B (seam 385/2), bright coal |
| | 11B | Bogdanka, Westphalian B (seam 385/2), dull coal |
| | 12B | Wesoła, Westphalian B (seam 308), bright coal |
| | 13B | Wesoła, Westphalian B (seam 308), dull coal |
| | 14B | Bielszowice, Upper Namurian (seam 405/2), bright coal |
| | 15B | Bielszowice Upper Namurian (seam 405/2), dull coal |

Samples of various lithotypes of humic coal were analyzed. The location, age, and the lithotype of coal of individual samples are described in Table 1. The collected samples were subjected to proximate and ultimate analysis. The analysis was performed in an accredited laboratory (Central Measurement and Testing Laboratory in Jastrzębie-Zdrój) in accordance with applicable standards: total moisture [29], ash content [30], volatile matter content [31], sulfur content [32], gross calorific value [33], ash fusibility [34], and carbon and hydrogen content [35].

Petrographic analysis of maceral groups of coal and random reflectance of ulminite/collotelinite was carried out in accordance with the following standards [36–38]. The analysis was carried out using a Zeiss microscope at the AGH University of Science and Technology in Kraków. The ash was obtained by burning coal samples at a temperature of 815 °C in a muffle furnace in accordance with ISO 1171 [30].

The analysis of the chemical composition of ash and the share of elements in coal and ash was conducted in an accredited laboratory (the Bureau Veritas Minerals Laboratory). ICP-MS analysis used a sample after modified aqua regia digestion (1:1:1 $HNO_3/HCl/H_2O$) for low to ultra-low determination of both coal and ash. The main parameters of the tested coal are presented in Table 2.

To supplement the petrographic and chemical analyses, XRD analysis was also performed. Qualitative and quantitative X-ray analyses of the samples were conducted using an APD diffractometer (Philips X'pert with a PW 3020 goniometer). Minerals were identified based on ICDD, ICSD, and COD databases.

**Table 2.** Proximate, ultimate and petrographic analysis of the examined coal.

| | Proximate Analysis | | | | | | | | | Ultimate Analysis | | | | | |
| No. | $M^{ad}$ (%) | $A^{db}$ (%) | $V^{daf}$ (%) | $S_t^{db}$ (%) | GCV (MJ/kg) | Ash Sintering Temperature ($t_S$) (°C) | Ash Softening Temperature ($t_A$) (°C) | Ash Melting Temperature ($t_B$)(°C) | Ash Fluid Temperature ($t_C$) (°C) | Total Porosity (%) | $C^{daf}$ (%) | $H^{daf}$ (%) | $N^{daf}$ (%) | $O^{daf}$ (%) | Aromaticity Factor (fa) |
|---|---|---|---|---|---|---|---|---|---|---|---|---|---|---|---|
| 1L | 7.6 | 46.2 | 56.2 | 1.6 | 23.9 | 890 | 1310 | 1400 | 1470 | 33.4 | 64.0 | 4.7 | 0.6 | 27.7 | 0.7 |
| 2L | 8.2 | 6.2 | 58.8 | 1.6 | 25.9 | 920 | 1220 | 1230 | 1240 | 25.0 | 64.9 | 5.3 | 0.2 | 27.8 | 0.6 |
| 3L | 9.6 | 24.6 | 56.9 | 2.8 | 26.1 | 990 | 1480 | 1510 | 1550 | 31.5 | 66.3 | 5.2 | 0.4 | 24.5 | 0.6 |
| 4L | 8.4 | 5.0 | 57.4 | 1.9 | 28.9 | 1110 | 1270 | 1310 | 1350 | 11.5 | 69.2 | 5.8 | 0.4 | 22.5 | 0.6 |
| 5L | 10.0 | 3.8 | 56.8 | 0.8 | 29.2 | 900 | 1210 | 1280 | 1320 | 26.5 | 70.7 | 6.0 | 0.5 | 22.0 | 0.6 |
| 6L | 10.5 | 6.9 | 55.8 | 1.0 | 26.2 | 920 | 1190 | 1290 | 1300 | 35.9 | 66.0 | 5.2 | 0.5 | 27.2 | 0.7 |
| 7L | 9.3 | 6.5 | 58.0 | 0.5 | 26.3 | 1020 | 1250 | 1280 | 1290 | 22.9 | 66.5 | 5.3 | 0.8 | 26.9 | 0.6 |
| 8L | 9.8 | 16.2 | 55.8 | 1.7 | 25.6 | 980 | 1230 | 1250 | 1260 | 8.4 | 65.7 | 5.0 | 0.7 | 26.6 | 0.7 |
| 9L | 8.7 | 7.2 | 56.2 | 3.6 | 26.9 | nd | nd | nd | nd | 19.9 | 63.4 | 5.2 | 0.2 | 27.3 | 0.7 |
| 10L | 12.0 | 19.8 | 55.4 | 3.6 | 25.5 | 910 | 1260 | 1280 | 1300 | 18.3 | 65.4 | 4.8 | 1.0 | 24.4 | 0.7 |
| 11L | 11.6 | 2.7 | 64.2 | 1.0 | 26.7 | 910 | 1300 | 1310 | 1320 | 28.6 | 65.0 | 5.8 | 0.1 | 28.1 | 0.5 |
| 12L | 4.6 | 23.2 | 49.4 | 3.9 | 27.6 | 980 | 1390 | 1430 | 1450 | 25.9 | 67.4 | 5.2 | 0.9 | 21.5 | 0.7 |
| 1SB | 9.7 | 22.4 | 58.3 | 3.6 | 25.2 | nd | nd | nd | nd | 11.1 | 65.1 | 5.0 | 0.8 | 24.6 | 0.6 |
| 1B | 0.6 | 6.5 | 22.0 | 1.1 | 36.4 | 960 | 1540 | 1550 | 1550 | 3.6 | 89.7 | 4.8 | 1.1 | 3.2 | 0.8 |
| 2B | 5.1 | 8.6 | 39.3 | 2.6 | 31.2 | 1090 | 1260 | 1340 | 1430 | 10.5 | 76.7 | 4.8 | 1.2 | 14.5 | 0.8 |
| 3B | 4.6 | 6.2 | 40.5 | 1.6 | 31.8 | 990 | 1240 | 1280 | 1380 | 9.2 | 77.9 | 5.1 | 1.3 | 14.0 | 0.7 |
| 4B | 5.1 | 7.8 | 35.4 | 1.4 | 31.2 | 900 | 1230 | 1270 | 1360 | 7.1 | 77.6 | 4.7 | 1.2 | 15.1 | 0.8 |
| 5B | 3.8 | 24.1 | 38.8 | 1.0 | 30.5 | 950 | 1550 | 1550 | 1550 | 6.1 | 76.0 | 5.1 | 1.2 | 16.4 | 0.8 |
| 6B | 1.5 | 8.3 | 39.5 | 3.0 | 33.4 | 900 | 1350 | 1510 | 1550 | 2.9 | 80.4 | 5.3 | 2.0 | 9.0 | 0.7 |
| 7B | 1.2 | 11.3 | 39.8 | 5.3 | 33.6 | 920 | 1330 | 1400 | 1410 | 2.6 | 79.4 | 5.4 | 1.8 | 7.5 | 0.7 |
| 8B | 1.1 | 3.3 | 35.1 | 1.2 | 33.9 | 940 | 1330 | 1380 | 1430 | 3.3 | 81.9 | 5.2 | 2.0 | 9.6 | 0.8 |
| 9B | 1.2 | 3.7 | 35.2 | 1.5 | 33.4 | 910 | 1340 | 1420 | 1440 | 3.5 | 81.1 | 5.2 | 2.0 | 10.1 | 0.8 |
| 10B | 1.1 | 6.0 | 38.0 | 1.3 | 33.9 | 830 | 1320 | 1450 | 1470 | 3.3 | 81.2 | 5.4 | 1.7 | 10.3 | 0.7 |
| 11B | 1.0 | 10.6 | 37.8 | 1.2 | 33.6 | 930 | 1540 | 1550 | 1550 | 3.8 | 81.7 | 5.3 | 1.5 | 10.1 | 0.7 |
| 12B | 2.6 | 10.4 | 40.0 | 0.9 | 29.0 | 890 | 1250 | 1300 | 1370 | 3.1 | 74.1 | 4.4 | 1.3 | 19.2 | 0.8 |
| 13B | 2.5 | 15.1 | 38.2 | 0.7 | 29.7 | 940 | 1270 | 1380 | 1420 | 4.0 | 74.1 | 4.5 | 1.3 | 19.3 | 0.8 |
| 14B | 0.6 | 10.5 | 32.1 | 0.4 | 35.0 | 930 | 1310 | 1340 | 1420 | 2.3 | 85.2 | 5.1 | 1.4 | 7.9 | 0.8 |
| 15B | 0.6 | 10.1 | 33.7 | 0.3 | 35.3 | 910 | 1260 | 1310 | 1380 | 2.3 | 85.5 | 5.3 | 1.4 | 7.4 | 0.8 |

nd—no data. ad—air dried. db—dry basis. daf—dry, ash free basis.

Based on chemical analysis and the Clarke value in the Earth's crust, the enrichment factor (EF) was calculated as the quotient of the element content and the Clarke value.

Statistical analysis was performed using Statistica software. Pearson's linear correlation coefficients (r) were calculated. The significance of this factor was assessed and analyzed. The test of statistical significance was performed using Student's t distribution (with $\alpha$ equal to 0.05). The strength of the correlation coefficient was determined on the basis of [39], where the value of the correlation coefficient can be interpreted as follows:

(0.00–0.20)—Weak correlation, the relationship is almost insignificant;
(<0.20–0.40)—Low correlation, the relationship is clearly visible;
(<0.40–0.70)—Strong correlation, the relationship is clearly visible and small;
(<0.70–0.90)—Strong correlation, significant relationship;
(<0.90–1.00)—Strong correlation and relationship.

## 3. Results and Discussion

### 3.1. Proximate and Ultimate Analysis

Basic technological and chemical parameters were analyzed in all samples. The main parameters of the tested coal are presented in Table 2. The tested parameters change depending on the rank of coal of the tested samples. The ash content in the dry basis ($A^{db}$) ranges from 2.7% to 46.2% and is clearly higher than in the case of lignite. The volatile matter content ($V^{daf}$) decreases with increasing rank of coal and ranges from 64.19% in xylitic coal from the Szczerców deposit to 21.95% in coking coal from the Jas-Mos deposit. A noticeable decrease in the total porosity of lignite and bituminous coal (35.91% and 2.27%, respectively) was also observed. On the other hand the gross calorific value ($GCV^{daf}$) increases from 23.91 MJ/kg to 36.41 MJ/kg with increasing rank of coal. In the tested samples, the carbon content ($C^{daf}$) ranges from 63.4% (lignite from the Pątnów deposit) to 89.73% (coking coal). In addition, ash melting points are not related to the rank of coal, but to the chemical composition of the ash. At the same time, the ash sintering, softening, melting and fluid temperatures of the tested samples are highly variable.

### 3.2. Petrographic Analysis of the Examined Coal

In terms of lithology, Polish lignite deposits contain mainly humic coal. Sapropelic and semi-sapropelic coals can be found in smaller quantities and, as of now, are not selectively exploited. Generally, Polish deposits are dominated by xylodetritic and detritic coal [40]. The bituminous coal deposits are dominated by humic coals with local sapropelic coal interlayers [41]. Variations of coal lithotypes were analyzed in order to determine the impact of petrographic composition on the chemical composition of ash and the content of selected elements. The huminite/vitrinite reflectance coefficient for the analyzed coal samples ranged from 0.24% in the Pątnów deposit to 1.07% in the Jas-Mos deposit (Table 3). The petrographic composition of lignite was dominated by huminite group macerals ranging from 73.6% to 97.7%. The composition depended on the lithotype of coal. The xylitic coal was dominated by textinite and ulminite; detritic coal was dominated by atrinite and densinite. The maximum content of inertinite macerals in lignite was 19.9% in the xylitic coal from the Pątnów deposit. However, the inertinite content generally did not exceed 7% (Table 3), which is a characteristic feature of Polish lignites [42]. The inertinite group in lignite was represented mainly by fusinite and inertodetrinite. The liptinite group in lignite was mainly resinite, sporinite, and liptodetrinite. The maximum content of liptinite macerals in the analyzed samples was 19.9% in the xylitic coal from the Pątnów deposit. The mineral matter content in the tested samples was highly variable and ranged from 0.0% to 11.1%. Generally, the mineral matter content in xylitic coal was low. Quartz and clay minerals were the most commonly observed in lignite. These minerals are more commonly found in detritic coal. Clay minerals were often mixed with atrinite. In addition, sulfides, mainly in the form of framboidal pyrite and pyrite impregnating textinite, were found in some of the lignite samples. Carbonates in the form of calcite and aragonite were observed in coal from the Bełchatów deposit.

The following lithotypes of bituminous coal were analyzed: dull coal and bright coal. The petrographic composition of the tested samples was more variable than that of lignite. The vitrinite macerals, dominated by collotelinite and collodetrinite, ranged from 40% to 85% (Table 3). In the case of dull coal, the vitrinite content did not exceed 55.4%, but the liptinite content was higher. The liptinite macerals in the tested samples were dominated by microsporinite, followed by macrosporinite and liptodetrinite. In the analyzed bright coal, the content of inertinite macerals was generally lower than in the case of dull coal. The inertinite content ranged from 4.8% to 38.2%; the most commonly observed macerals of this group were fusinite, semifusinite, funginite, macrinite, and inertodetrinite. The mineral matter content of bituminous coal ranged from 0.0% to 7.6%. The most common minerals were pyrite, marcasite and clay minerals. In the sample from Bielszowice, carbonates were also determined (Table 3).

<p style="text-align:center">**Table 3.** Petrographic analysis of the examined coal.</p>

| No. | Huminite Vitrinite (%) | Inertinite (%) | Liptinite (%) | Mineral Matter (%) | Sulfides | Carbonates | Quartz + Clays | Random Reflectance (Ro) (%) |
|---|---|---|---|---|---|---|---|---|
| 1L | 90.6 | 2.5 | 2.3 | 4.6 | 1.0 | 0.0 | 3.6 | 0.24 |
| 2L | 74.8 | 19.9 | 1.7 | 3.6 | 0.0 | 0.0 | 3.6 | 0.24 |
| 3L | 89.6 | 3.8 | 2.0 | 4.6 | 0.0 | 0.0 | 4.6 | 0.24 |
| 4L | 94.0 | 0.3 | 2.7 | 3.0 | 0.0 | 0.0 | 3.0 | 0.28 |
| 5L | 94.5 | 0.8 | 3.4 | 1.3 | 0.0 | 0.0 | 1.3 | 0.28 |
| 6L | 93.7 | 0.6 | 2.9 | 2.7 | 0.0 | 0.0 | 2.7 | 0.25 |
| 7L | 84.8 | 1.6 | 4.0 | 9.6 | 0.8 | 0.0 | 8.8 | 0.25 |
| 8L | 89.9 | 1.9 | 2.7 | 5.6 | 0.4 | 0.4 | 4.8 | 0.28 |
| 9L | 95.8 | 1.5 | 1.8 | 1.0 | 0.0 | 0.0 | 1.0 | 0.28 |
| 10L | 92.7 | 1.6 | 2.6 | 3.1 | 1.6 | 0.0 | 1.6 | 0.28 |
| 11L | 97.1 | 0.8 | 1.7 | 0.4 | 0.0 | 0.0 | 0.4 | 0.26 |
| 12L | 73.6 | 6.8 | 8.5 | 11.1 | 0.8 | 0.0 | 10.4 | 0.26 |
| 1SB | 83.0 | 3.2 | 8.4 | 5.2 | 0.1 | 0.0 | 0.3 | 0.36 |
| 1B | 76.5 | 23.0 | 0.0 | 0.5 | 0.5 | 0.0 | 0.0 | 1.07 |
| 2B | 77.8 | 13.2 | 7.8 | 1.2 | 0.6 | 0.0 | 0.6 | 0.51 |
| 3B | 40.4 | 30.4 | 28.2 | 1.0 | 0.8 | 0.0 | 0.2 | 0.51 |
| 4B | 68.8 | 24.4 | 5.2 | 1.6 | 0.6 | 0.0 | 1.0 | 0.54 |
| 5B | 55.4 | 23.2 | 13.8 | 7.6 | 1.4 | 0.0 | 6.2 | 0.54 |
| 6B | 85.0 | 4.8 | 7.6 | 2.6 | 2.0 | 0.0 | 0.6 | 0.62 |
| 7B | 47.6 | 20.8 | 27.4 | 4.2 | 3.8 | 0.0 | 0.4 | 0.62 |
| 8B | 80.8 | 9.0 | 9.6 | 0.6 | 0.0 | 0.0 | 0.6 | 0.59 |
| 9B | 54.0 | 19.4 | 26.0 | 0.6 | 0.4 | 0.0 | 0.2 | 0.59 |
| 10B | 75.2 | 10.2 | 13.4 | 1.2 | 0.8 | 0.0 | 0.4 | 0.61 |
| 11B | 46.8 | 24.6 | 27.0 | 1.6 | 0.2 | 0.0 | 1.4 | 0.61 |
| 12B | 78.8 | 8.4 | 9.6 | 3.2 | 0.0 | 0.4 | 2.8 | 0.63 |
| 13B | 40.0 | 31.2 | 26.8 | 2.0 | 0.4 | 0.0 | 1.6 | 0.63 |
| 14B | 69.2 | 22.6 | 8.2 | 0.0 | 0.0 | 0.0 | 0.0 | 0.70 |
| 15B | 42.0 | 38.2 | 18.6 | 1.2 | 0.0 | 0.0 | 1.2 | 0.70 |

### 3.3. Oxide Composition of Ash and Its Properties

Analysis of the chemical composition of ash from coal is essential when determining the possible uses of coal for a given energy technology. In addition, it allows the use of residues from the combustion or gasification of coal as a material for building or ceramics.

Based on the XDR analysis, it was found that the mineral matter composition was slightly different for lignite and bituminous coal. In the case of lignite, the main components were quartz and clay minerals (illite–kaolinite–smectite), followed by iron sulfides in the form of pyrite and marcasite, chlorites, plagioclases, potassium feldspars, gypsum, and glauconite. Carbonates, mainly as lake chalk (calcite and dolomite), occur in the Bełchatów and Szczerców deposits. Bituminous coal contains less quartz than lignite; the most commonly observed are clay minerals: kaolinite and illite. Sulfides occur in the form of pyrites, marcasites, galena, and sphalerite. The carbonates in the examined deposits included calcite and dolomite. Some samples contained siderite. The chemical composition of ash differs significantly from the original mineral matter, which undergoes significant changes during the combustion (oxidation) [43–46], such as the following:

- Carbonates decompose with the release of carbon dioxide: $CaCO_3 \rightarrow CaO + CO_2$;
- Silicates and aluminosilicates lose crystallization water;
- Pyrite and marcasite oxidize to $Fe_2O_3$ and $SO_2$: $4FeS_2 + 11O_2 \rightarrow 2Fe_2O_3 + 8SO_2$;
- Calcium carbonate, when broken down into calcium oxide, can bind $SO_2$ resulting from the oxidation of pyrites and organic sulfur to calcium sulfate (IV): $CaO + SO_2 \rightarrow CaSO_3$, which then further oxidizes to calcium (VI) sulfate: $2CaSO_3 + O_2 \rightarrow 2CaSO_4$;
- Alkali metal chlorides sublime (volatilize, e.g., NaCl);
- Oxidation of organometallic compounds.

These reactions occur at temperatures up to about 1070 K (800 °C); therefore, in order to obtain a stabilized ash mass, a combustion temperature of 1088 ± 10 K (815 °C ± 10 °C) and a sufficiently long

annealing time are recommended. The amount of ash is usually lower than the original mineral matter content. The content and the type of mineral matter in coal also have an impact on the combustion process. The currently used methods allow the combustion of coal with a dry ash content below 40%. According to Shirazi et al. [47], the amount and chemical composition of mineral matter significantly reduces the amount of obtained energy. This is due to the fact that part of it is used for heating the non-combustible material. It was found that the energy loss was more significant for pulverized coal-fired boilers. Ash components react with each other at high temperatures and—creating mineral phases and volatile compounds—affect the combustion process [48–50]. Kaolinite breaks up at a temperature of 450 °C, losing OH units in the crystal structure. In turn, other clay minerals (mostly illite and smectite) decompose at about 500 °C, forming new mineral phases which can lead to the plasticization of ash and formation of a difficult-to-remove slag. Calcite and dolomite break out at a temperature of about 900 °C, contributing to the formation of calcium oxide (CaO), which reacts with water vapor to form a mineral phase composed of artificial minerals similar to portlandite ($Ca(OH)_2$). The reaction with aluminosilicates results in the formation of gehlenite ($Ca_2Al_2SiO_7$) and anorthite ($CaAl_2Si_2O_8$). Carbonates, usually calcite, can react with the sulfur in the coal and form an artificial mineral phase with the chemical composition of anhydrite, and thus bind sulfur [10]. This process has been widely used as one of the methods for desulfurization of dry flue gas generated during combustion. The change in mineral composition was clearly visible in the results of the XDR analysis, where the most common components are quartz, illite, hematite, anhydrite, microcline, and magnetite.

The chemical composition of ashes obtained as a result of slow combustion of coal is shown in Table 4. Figure 1 presents the chemical composition of the tested coal recalculated to 100% without ignition losses. It has been shown that the oxide composition of bituminous coal and lignite differs significantly. In lignite, $SiO_2$, CaO, $Fe_2O_3$, and $Al_2O_3$ are the dominant components. Interestingly, during the analysis of the oxide composition of lignite, relatively high ignition losses were also observed. Ignition losses reached up to 38.9% (detritic lignite from the Sieniawa deposit). Such high ignition losses are due to the presence of organic matter in the unburned ash [46,51]. It is mainly degassed char or coke breeze. With reference to the analyses presented in Table 4 and Figure 1, it should be noted that they represent the chemical composition instead of mineral composition and the traditional assumption that all components are in the form of oxides. What distinguishes lignite ash from bituminous coal ash is a high CaO content. In particular, this concentration is significant in the Szczerców, Bełchatów, and Sieniawa fields. The CaO content in the tested ashes reached 30% in the Sieniawa deposit. The 25% CaO content in the Bełchatów and Szczerców deposits is related to the lake chalk content. Generally, the calcium content of coal is expressed as CaO content. It should be noted that the CaO content in the oxide analysis does not mean that there was actually CaO in the ash or in the coal. The calcium content of coal was associated with mineral matter found in coal, mainly calcite ($CaCO_3$), partly aragonite ($CaCO_3$), and rarely occurring calcium and aluminum hydroxycarbonates, e.g., para-alumohydrocalcite ($CaAl_2(CO_3)_2(OH)_4 \cdot 6H_2O$) [52,53]. However, it should be noted that part of the calcium, together with Fe, Mg, K and Na, forms, as a result of reaction with organic matter (coal), dopplerites [54]. Calcium oxide in bituminous coal ashes is present in smaller amounts, generally up to 10%. The exception was vitrinite coal ash from the No. 308 seam of the Wesoła deposit.

**Table 4.** Chemical composition of coal ash.

| No. | SiO$_2$ (%) | Al$_2$O$_3$ (%) | Fe$_2$O$_3$ (%) | MgO (%) | CaO (%) | Na$_2$O (%) | K$_2$O (%) | TiO$_2$ (%) | P$_2$O$_5$ (%) | MnO (%) | Cr$_2$O$_3$ (%) | LOI and Others (%) | C in Ash (%) | S in Ash (%) |
|---|---|---|---|---|---|---|---|---|---|---|---|---|---|---|
| 1L | 66.8 | 16.9 | 2.2 | 1.6 | 6.1 | 0.1 | 0.6 | 1.7 | 0.0 | 0.1 | 0.2 | 3.8 | 0.1 | 1.4 |
| 2L | 37.4 | 9.9 | 7.8 | 3.5 | 18.1 | 0.3 | 0.6 | 0.6 | 0.3 | 0.1 | 0.1 | 21.2 | 0.2 | 8.7 |
| 3L | 64.3 | 3.0 | 6.3 | 2.3 | 9.4 | 0.1 | 0.2 | 0.5 | 0.0 | 0.2 | 0.2 | 13.6 | 0.1 | 5.7 |
| 4L | 19.3 | 15.0 | 21.9 | 6.4 | 4.9 | 7.8 | 1.2 | 1.6 | 0.2 | 0.0 | 0.1 | 21.5 | 0.3 | 6.5 |
| 5L | 18.2 | 19.0 | 6.2 | 11.0 | 8.4 | 12.3 | 1.0 | 2.4 | 0.2 | 0.0 | 0.1 | 21.2 | 0.8 | 6.7 |
| 6L | 20.4 | 3.5 | 11.7 | 2.4 | 26.1 | 0.3 | 0.3 | 0.3 | 0.0 | 0.1 | 0.0 | 34.9 | 2.3 | 9.6 |
| 7L | 7.4 | 3.2 | 17.6 | 2.1 | 30.0 | 0.1 | 0.1 | 0.3 | 0.0 | 0.2 | 0.0 | 38.9 | 3.4 | 8.3 |
| 8L | 25.5 | 16.7 | 7.2 | 2.2 | 23.6 | 0.5 | 0.2 | 0.6 | 0.2 | 0.2 | 0.0 | 23.1 | 0.8 | 7.8 |
| 9L | 19.7 | 13.8 | 8.6 | 1.6 | 24.4 | 0.1 | 0.4 | 0.8 | 0.2 | 0.1 | 0.1 | 30.3 | 0.2 | 12.5 |
| 10L | 20.4 | 13.9 | 9.7 | 1.3 | 24.5 | 0.0 | 0.2 | 0.7 | 0.1 | 0.1 | 0.0 | 29.1 | 0.2 | 11.7 |
| 11L | 18.4 | 7.4 | 7.6 | 2.2 | 25.6 | 0.2 | 0.4 | 0.5 | 0.1 | 0.1 | 0.1 | 37.6 | 0.6 | 13.2 |
| 12L | 63.4 | 11.8 | 16.4 | 0.3 | 1.2 | 0.1 | 0.8 | 2.9 | 0.1 | 0.0 | 0.0 | 2.9 | 0.1 | 0.5 |
| 1SB | 28.2 | 14.5 | 9.2 | 1.1 | 20.6 | 0.1 | 0.1 | 0.8 | 0.1 | 0.1 | 0.0 | 25.1 | 0.2 | 10.2 |
| 1B | 45.1 | 35.0 | 14.0 | 0.7 | 1.4 | 0.9 | 0.8 | 1.4 | 0.2 | 0.0 | 0.1 | 0.6 | 0.2 | 3.3 |
| 2B | 25.6 | 20.7 | 29.9 | 3.1 | 7.0 | 5.3 | 1.0 | 0.6 | 0.5 | 0.1 | 0.1 | 6.2 | 0.1 | 1.8 |
| 3B | 31.2 | 23.8 | 19.5 | 2.5 | 6.0 | 6.3 | 0.5 | 0.8 | 0.3 | 0.0 | 0.0 | 9.1 | 0.1 | 0.5 |
| 4B | 35.1 | 27.9 | 14.9 | 2.4 | 4.2 | 6.0 | 1.7 | 1.0 | 0.2 | 0.0 | 0.1 | 6.7 | 0.6 | 8.6 |
| 5B | 50.7 | 32.1 | 4.9 | 1.3 | 1.0 | 1.7 | 3.1 | 1.2 | 0.1 | 0.0 | 0.0 | 3.9 | 0.6 | 7.6 |
| 6B | 9.6 | 8.7 | 58.9 | 2.8 | 5.2 | 0.4 | 0.2 | 0.2 | 1.2 | 0.6 | 0.1 | 12.1 | 0.2 | 1.9 |
| 7B | 24.0 | 21.1 | 46.6 | 0.6 | 1.1 | 0.4 | 0.7 | 0.7 | 1.3 | 0.1 | 0.1 | 3.3 | 0.2 | 0.3 |
| 8B | 38.4 | 31.7 | 15.3 | 2.3 | 4.9 | 1.0 | 2.0 | 1.5 | 1.3 | 0.1 | 0.1 | 1.4 | 0.4 | 1.6 |
| 9B | 38.9 | 31.9 | 16.8 | 2.0 | 3.9 | 0.7 | 2.0 | 1.2 | 1.1 | 0.1 | 0.1 | 1.3 | 0.5 | 1.1 |
| 10B | 36.3 | 32.8 | 15.8 | 0.5 | 6.3 | 0.5 | 0.5 | 1.2 | 4.6 | 0.0 | 0.1 | 1.4 | 0.2 | 0.4 |
| 11B | 43.3 | 37.9 | 8.4 | 0.4 | 2.5 | 0.4 | 0.4 | 1.5 | 2.1 | 0.0 | 0.0 | 3.0 | 0.2 | 0.3 |
| 12B | 16.3 | 15.3 | 11.5 | 8.8 | 19.7 | 1.0 | 0.5 | 0.7 | 3.1 | 0.4 | 0.0 | 22.8 | 1.5 | 4.5 |
| 13B | 41.9 | 29.9 | 5.8 | 2.3 | 6.2 | 1.1 | 1.5 | 1.5 | 3.8 | 0.1 | 0.0 | 5.8 | 0.3 | 1.3 |
| 14B | 39.8 | 29.8 | 3.9 | 3.6 | 8.0 | 2.0 | 1.2 | 1.2 | 2.6 | 0.0 | 0.0 | 8.0 | 0.3 | 0.4 |
| 15B | 39.3 | 29.5 | 3.6 | 4.0 | 9.3 | 1.9 | 1.1 | 1.2 | 3.4 | 0.1 | 0.0 | 6.6 | 0.7 | 1.9 |

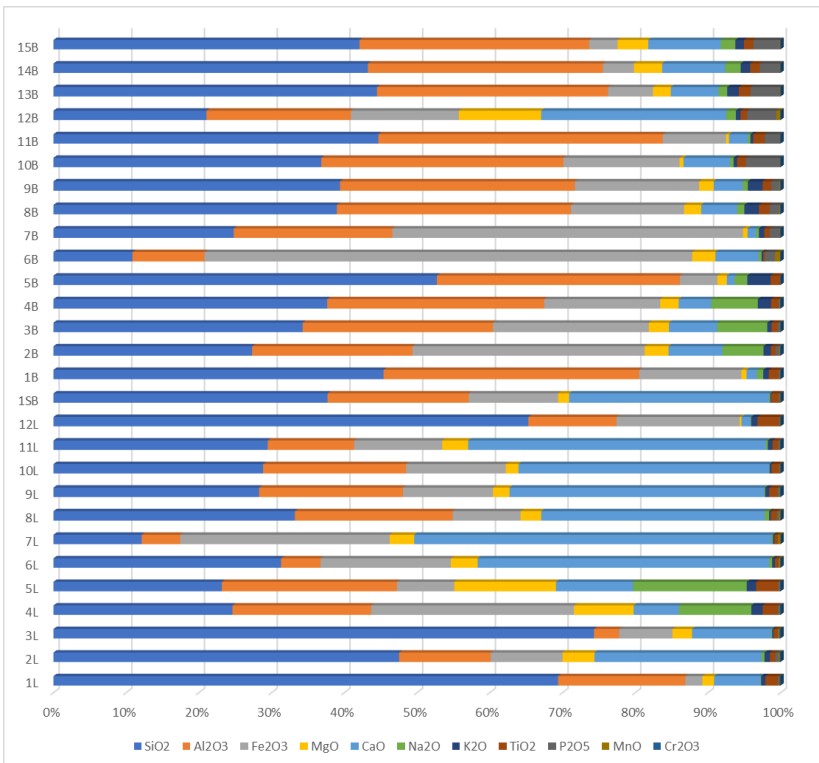

**Figure 1.** Chemical composition of ashes in the tested samples recalculated to 100% without ignition losses.

Silica (silicon dioxide, $SiO_2$), which in some lignite deposits constitutes over 60% of ash, is an important component of ash. Detritic coal ashes have a high $SiO_2$ content. $SiO_2$ is bound by silicates and aluminosilicates, mainly quartz and clay minerals. These minerals are quite common in deposits and are associated with their genesis. The increased $SiO_2$ content in ash suggests the occurrence of alluvial fans in coal deposits [55] or high detrital input ratios and/or sediment-laden water influences into palaeomires [46,53,56,57]. The $SiO_2$ content in bituminous coal ash from the Janina deposit was up to 50%.

The $Al_2O_3$ content in the tested samples was clearly higher in bituminous coal ashes. This content reached 38% in the Bogdanka deposit. In the case of lignite ash, the maximum $Al_2O_3$ content of up to 19% was found in detritic coal from the Turów deposit. In addition, the ash had an anomalously high content of MgO, $Na_2O$, $K_2O$ and $TiO_2$ compared to other tested samples. The share of $Fe_2O_3$ in the analyzed ashes was highly variable and ranged from 2.2% to 58.9%. It should be noted that the ash from the No. 391 seam in the Bogdanka deposit contained about 50% of $Fe_2O_3$. The increased content of $Fe_2O_3$ is associated with a significant amount of pyrite and siderite in coal.

Based on the analysis of Pearson's linear correlation, it was found that the ash oxide composition affects some technological parameters of coal (Appendix A). Usually these correlations are either moderate or strong. A negative strong correlation between the volatile matter content ($V^{daf}$) and the amount of $Al_2O_3$ in the ash (Figure 2) and a moderate negative correlation between $V^{daf}$ and $K_2O$ and $P_2O_5$ were found. Additionally, an increase in CaO in ash increased the volatile content of coal. Inverse relationships to those observed for $V^{daf}$ were demonstrated for gross calorific value (GCV) and carbon ($C^{daf}$) content in coal. In these cases, the increase in $Al_2O_3$ in the ash strongly positively affected GCV (Figure 3) and $C^{daf}$ (Figure 4). Meanwhile, GCV and $C^{daf}$ in coal decreased with the increase in the CaO content. It should be borne in mind that the correlation does not imply causation, but it shows some dependencies.

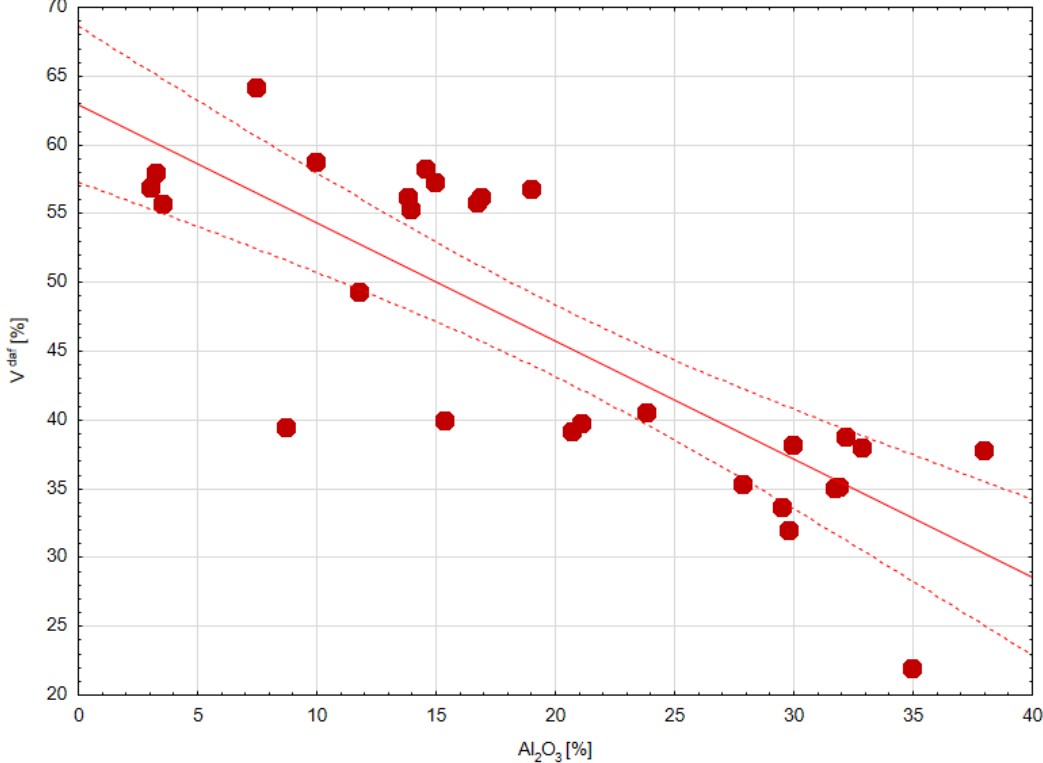

**Figure 2.** Correlation between volatile matter ($V^{daf}$) and $Al_2O_3$ content in ash.

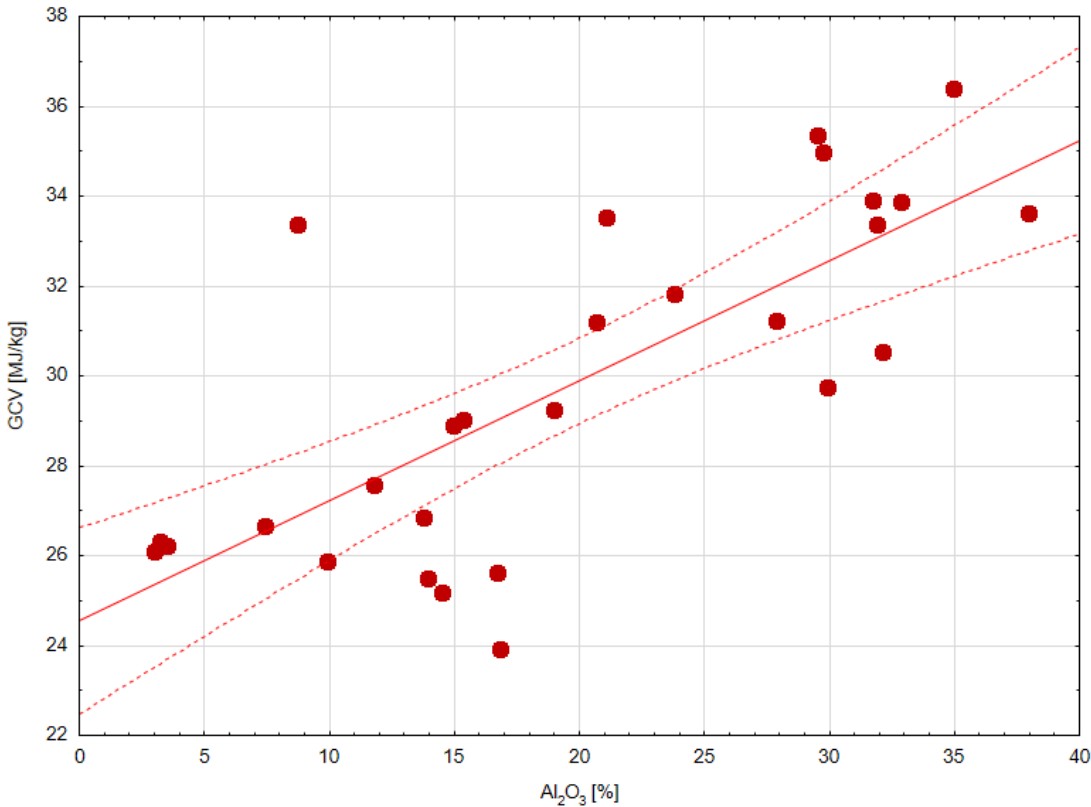

**Figure 3.** Correlation between gross calorific value (GCV) and $Al_2O_3$ content in ash.

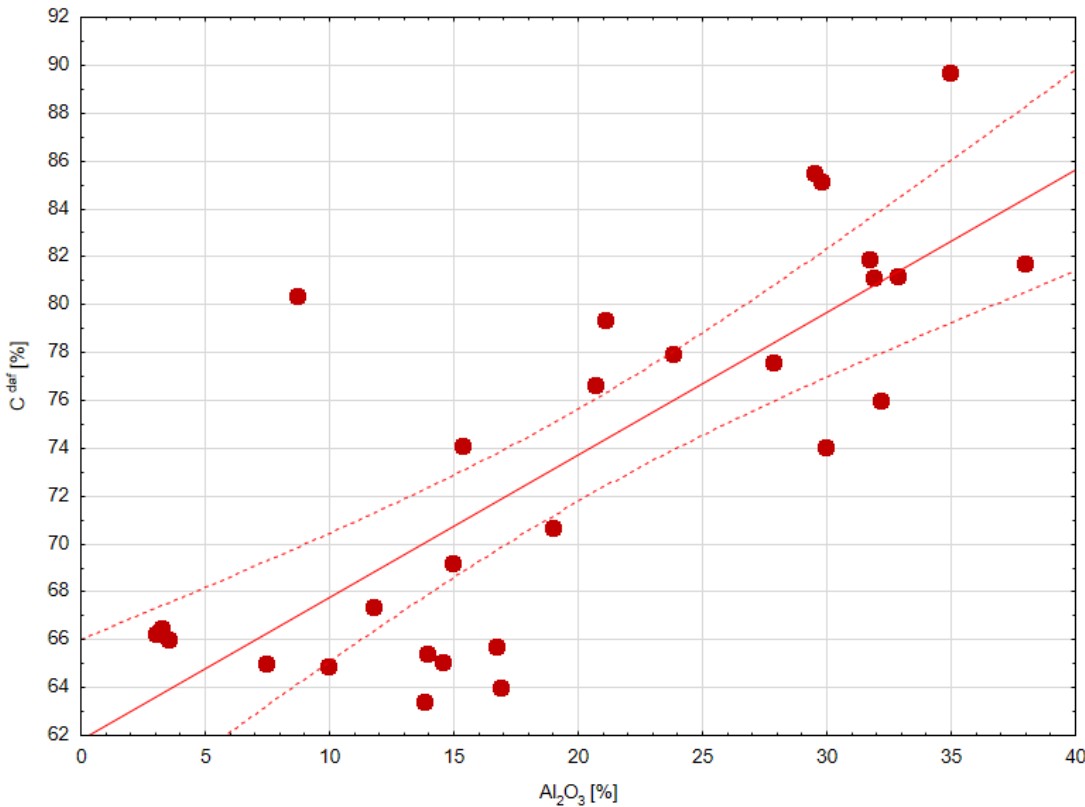

**Figure 4.** Correlation between carbon content in coal ($C^{daf}$) and $Al_2O_3$ content in ash.

The ash melting point is of great importance when estimating the influence of mineral matter on a selected combustion method. Individual minerals have different melting points. The low ash melting point is the reason for many unfavorable phenomena during combustion processes. The vitrified ash damages boiler and gas generator grates and their housing material, hinders the air circulation and sticks to the unburned coal, which results in a decreased thermal efficiency of the furnace [58,59]. As a result of reacting with the material of the furnace walls, it can create easily fusible slag contributing to serious accidents. The ash melting point depends on its chemical composition. The more aluminosilicates and the less oxide combinations of iron and alkali, the higher the mentioned temperature [60,61]. The phenomena of ash softening and melting have a very complex nature. They run in parallel with chemical reactions, which are greatly affected by the duration of the process, fragmentation, and distribution of ash components. Both before and during melting, the ash components react with each other, as well as with their degradation products. Due to the fact that quartz has a higher melting point (approximately 1800 °C) compared to furnace temperatures [50], it does not go into a liquid state and is thus the least problematic. Based on the correlation, a moderate effect of $SiO_2$, CaO, and $P_2O_5$ content on softening ($t_A$), melting ($t_B$) and fluid ($t_C$) points was confirmed.

When evaluating the usefulness of coal for energy generation via combustion and gasification, it is necessary to predict the behavior of ash at high temperatures and its impact on the process. The ash fouling and slagging processes, which are dependent on the ash content, composition, melting point and on the chlorine, phosphorus, and sulfur content in coal, have a negative impact on the combustion process [62]. To determine the tendency to form unfavorable residues in the furnace, the following indices [63,64] are determined:

(1) alkalinity/acidity (base/acid (B/A) ratio)

$$B/A = \%(Fe_2O_3 + CaO + MgO + Na_2O + K_2O)/[\%(SiO_2 + Al_2O_3 + TiO_2)] \quad (1)$$

where $SiO_2$, $Al_2O_3$, $TiO_3$, $Fe_2O_3$, CaO, MgO, $Na_2O$, and $K_2O$ represent the percentages of individual oxides in the ashes.

(2) Slagging (slagging index) (Rs)

$$Rs = B/A \times S_t{}^d \quad (2)$$

where $S_t{}^d$ is the total sulfur content in coal, dry basis. Classification of slagging potential using Slagging Index Rs

<0.6 low
≤0.6–2.0 medium
2.0–2.6 high
>2.6 severe

(3) $SiO_2$ ratio (silica value, SV)

$$SV = (SiO_2 \times 100)/(SiO_2 + Fe_2O_3 + CaO + MgO) \quad (3)$$

Depending on the silica value SV, the coal will show

≥72 low tendency to slagging
65–72 medium tendency to slagging
≤65 high tendency to slagging

(4) The tendency to heating surfaces slagging (*Fouling Index Rf)*

$$Rf = B/A \times (Na_2O + K_2O) \quad (4)$$

Classification of fouling potential using Fouling Index Rf

<0.6 low

0.6–40 high

>40.0 severe

(5)　　Alkalinity AK

$$AK = Na_2O + 0.96559K_2O \times (A^d/100) \tag{5}$$

where $A^d$ is the percentage of ash content in coal (dry basis). Alkali indexes AK:

<0.3 low tendency to pollute the alkaline components

0.3–0.45 medium tendency to pollute the alkaline components

0.45–0.6 high tendency to pollute the alkaline components

>0.6 severe tendency to pollute the alkaline components

According to the literature, the coals with chlorine content greater than 0.3%, phosphorus content over 0.03%, and sulfur content higher than 1.8% have the greatest tendency to fouling. According to Atakül et al. [65], it is directly proportional to the $Na_2O$ content. At the same time, it is inversely proportional to the content of $Al_2O_3$. The indexes for bituminous coal and lignite ashes are presented in Table 4.

Among the tested samples, the ash from the Bogdanka deposit shows the least favorable indicators, as severe slagging tendency and high alkalinity were observed. A very high ash slagging tendency was observed in ashes from the Bełchatów deposit. The correlation between the low slagging tendency and severe tendency to pollute the alkaline components was clearly visible in the Janina deposit. Coal from the Jóźwin deposit showed the most favorable ash parameters. This is related to the combustion process.

### 3.4. Critical Elements in Coal and Ash

This paper examines the content of selected critical elements: Ga, Sc, and V. The contents of selected elements were determined in coal and ash samples. These contents were compared with the Clarke values in the Earth's crust [66,67] and the Clarke values determined in coal and coal ashes [68]. The graphs show the determined contents for individual samples compared to the Clarke values.

### 3.4.1. Ga—Gallium

Elemental gallium is not found in nature, but can be obtained by smelting. It occurs in trace amounts in bauxite, kaolinite, and zinc ores. On an industrial scale, it is obtained from bauxite [69]. The demand for gallium will continue to increase over the next few years; this is related to new technologies such as fifth-generation wireless network technology (5G), solar cells, and the use of light-emitting diodes (LEDs) that replace incandescent and fluorescent lamps in lighting applications.

Fluoride, arsenide, and gallium phosphide have semiconductor properties. They are used, as admixtures to silicon, in the electronics industry. Gallium arsenide layers are used in monolithic microwave integrated circuits (MMICs) [70]. Gallium hydrides can also be used for hydrogen storage [71]. Gallium is mainly used:

-　　As an admixture in the production of semiconductors and transistors;
-　　In the production of mirrors (to moisten the glass);
-　　In the production of low-melting alloys;
-　　To improve the properties of solders;
-　　For the production of high temperature thermometers;
-　　As a catalyst in the production of hydrogen from water by aluminum oxidation;
-　　For science tricks [72] (e.g., the mind-bending gallium spoon trick, which is a classic magician trick [71]).

To sum up, gallium is one of the key elements in modern technologies, which are currently being intensively developed. This makes it one of the most intensively sought-after chemical elements.

In 2017, the world production of low-grade gallium was about 315 tons, the main producers being China, Japan, South Korea, Russia and Ukraine. The yield of high-purity gallium was about 180 tons, (produced mainly in China, Japan, Slovakia, the United Kingdom, and the United States) [73].

A search for alternative sources of gallium began at the start of the 21st century. In the first decade, high Ga contents in coal were recorded in China, Inner Mongolia [74–79]. Gallium, which is a rare metal, is anomalously enriched in coal to such an extent that it can be considered an economic resource. The mentioned scientific studies laid the foundations for the production of gallium from coal led to the development of a pilot plant in Inner Mongolia in China with a planned annual capacity of about 150 tons of Ga generated from coal combustion residues [80]. Most studies on Ga in coal and coal ash were focused on its origin and forms of occurrence. Recently, studies on gallium in coal have gone into attempts to recover Ga from residues after the combustion process (both at laboratory and industrial scales) [81].

The recovery of Ga from coal or by-products of coal combustion is a promising alternative way of obtaining this metal; such techniques are likely to lead to economic benefits and a positive environmental impact.

According to the literature, the Ga content in coal deposits can reach 263 ppm [82], with Clarke values for this element determined at 5.5 ppm in lignite, 6.0 ppm in bituminous coal, and 5.8 ppm on average [39]. In the case of coal ashes, the gallium Clarke value is 29 ppm for lignite ash, 36 ppm for bituminous coal ash and, on average, 33 ppm for coal ashes [39]. In the examined samples of Polish coal, the average Ga content was 2.3 ppm in lignite and 2.1 ppm in bituminous coal (Table 5). Thus, the Ga content in coal was lower than the Ga Clarke value in coal and in the Earth's crust (12 ppm). Ga contents in individual beds are depicted in Figure 5.

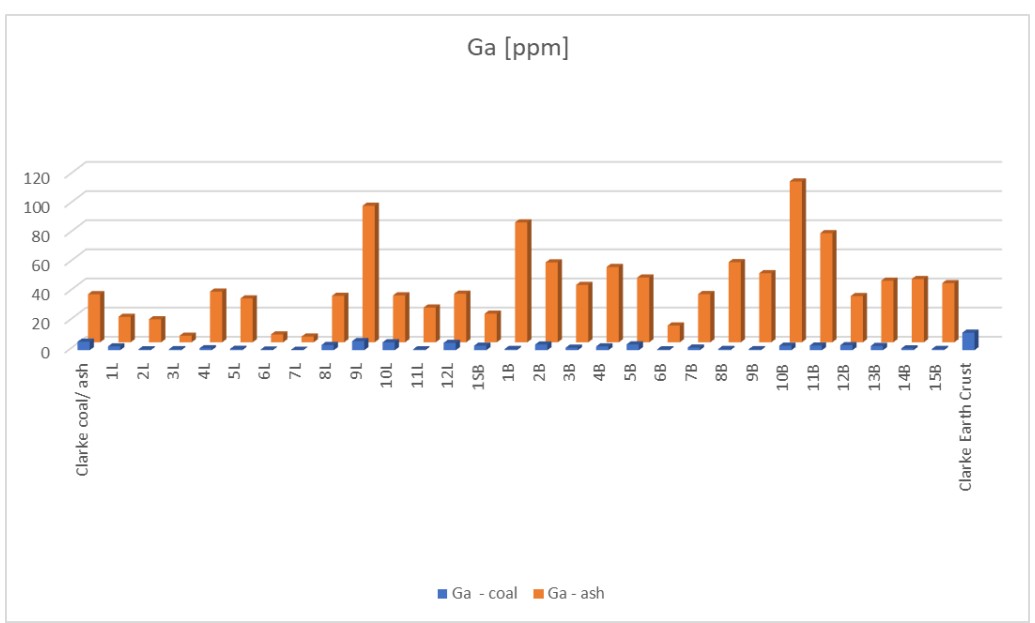

**Figure 5.** Ga content in coal and ash.

**Table 5.** The content of Ga, Sc and V in coal and ashes and enrichment factors against the Clarke value in the Earth's crust.

| No. | Ga (ppm) Coal | EF Ga Coal | Ga (ppm) Ash | EF Ga Ash | Sc (ppm) Coal | EF Sc Coal | Sc (ppm) Ash | EF Sc Ash | V (ppm) Coal | EF V Coal | V (ppm) Ash | EF V Ash |
|---|---|---|---|---|---|---|---|---|---|---|---|---|
| Average Clarke [49] | 5.8 | | 33.0 | | 3.9 | | 23.0 | | 25.0 | | 155.0 | |
| 1L | 2.6 | 0.2 | 17.5 | 1.5 | 1.3 | 0.1 | 6.0 | 0.3 | 12.0 | 0.1 | 70.0 | 0.6 |
| 2L | 0.6 | 0.1 | 15.8 | 1.3 | 0.6 | 0.0 | 10.0 | 0.5 | 11.0 | 0.1 | 231.0 | 1.9 |
| 3L | 0.7 | 0.1 | 4.5 | 0.4 | 0.7 | 0.0 | 3.0 | 0.1 | 6.0 | 0.1 | 27.0 | 0.2 |
| 4L | 1.3 | 0.1 | 34.8 | 2.9 | 3.4 | 0.2 | 62.0 | 2.8 | 69.0 | 0.6 | 1426.0 | 11.9 |
| 5L | 0.9 | 0.1 | 30.1 | 2.5 | 1.9 | 0.1 | 51.0 | 2.3 | 20.0 | 0.2 | 559.0 | 4.7 |
| 6L | 0.4 | 0.0 | 5.4 | 0.5 | 0.4 | 0.0 | 5.0 | 0.2 | 3.0 | 0.0 | 69.0 | 0.6 |
| 7L | 0.3 | 0.0 | 3.9 | 0.3 | 0.5 | 0.0 | 6.0 | 0.3 | 2.0 | 0.0 | 31.0 | 0.3 |
| 8L | 3.6 | 0.3 | 31.9 | 2.7 | 2.6 | 0.1 | 17.0 | 0.8 | 18.0 | 0.2 | 162.0 | 1.4 |
| 9L | 6.2 | 0.5 | 93.7 | 7.8 | 2.0 | 0.1 | 30.0 | 1.4 | 90.0 | 0.8 | 1404.0 | 11.7 |
| 10L | 5.4 | 0.5 | 32.2 | 2.7 | 6.4 | 0.3 | 35.0 | 1.6 | 58.0 | 0.5 | 363.0 | 3.0 |
| 11L | 0.6 | 0.1 | 23.8 | 2.0 | 0.4 | 0.0 | 12.0 | 0.5 | 4.0 | 0.0 | 178.0 | 1.5 |
| 12L | 5.0 | 0.4 | 33.3 | 2.8 | 8.0 | 0.4 | 43.0 | 2.0 | 25.0 | 0.2 | 176.0 | 1.5 |
| Average Lignite | 2.3 | 0.2 | 27.2 | 2.3 | 2.4 | 0.1 | 23.3 | 1.1 | 26.5 | 0.2 | 391.3 | 3.3 |
| 1SB | 3.1 | 0.3 | 19.6 | 1.6 | 3.7 | 0.2 | 18.0 | 0.8 | 37.0 | 0.3 | 226.0 | 1.9 |
| 1B | 0.8 | 0.1 | 82.3 | 6.9 | 0.9 | 0.0 | 64.0 | 2.9 | 11.0 | 0.1 | 626.0 | 5.2 |
| 2B | 3.9 | 0.3 | 54.8 | 4.6 | 2.2 | 0.1 | 32.0 | 1.5 | 17.0 | 0.1 | 277.0 | 2.3 |
| 3B | 1.7 | 0.1 | 39.4 | 3.3 | 1.8 | 0.1 | 32.0 | 1.5 | 12.0 | 0.1 | 227.0 | 1.9 |
| 4B | 2.6 | 0.2 | 51.7 | 4.3 | 2.2 | 0.1 | 33.0 | 1.5 | 37.0 | 0.3 | 554.0 | 4.6 |
| 5B | 4.0 | 0.3 | 44.4 | 3.7 | 4.1 | 0.2 | 29.0 | 1.3 | 44.0 | 0.4 | 285.0 | 2.4 |
| 6B | 0.6 | 0.1 | 11.5 | 1.0 | 0.6 | 0.0 | 10.0 | 0.5 | 15.0 | 0.1 | 210.0 | 1.8 |
| 7B | 1.8 | 0.2 | 33.1 | 2.8 | 2.0 | 0.1 | 38.0 | 1.7 | 45.0 | 0.4 | 539.0 | 4.5 |
| 8B | 0.8 | 0.1 | 55.0 | 4.6 | 0.7 | 0.0 | 45.0 | 2.0 | 26.0 | 0.2 | 1112.0 | 9.3 |
| 9B | 0.7 | 0.1 | 47.4 | 4.0 | 0.8 | 0.0 | 41.0 | 1.9 | 35.0 | 0.3 | 1277.0 | 10.6 |
| 10B | 3.1 | 0.3 | 110.4 | 9.2 | 2.0 | 0.1 | 61.0 | 2.8 | 56.0 | 0.5 | 1234.0 | 10.3 |
| 11B | 3.2 | 0.3 | 74.9 | 6.2 | 2.2 | 0.1 | 40.0 | 1.8 | 39.0 | 0.3 | 545.0 | 4.5 |
| 12B | 3.4 | 0.3 | 31.7 | 2.6 | 2.3 | 0.1 | 24.0 | 1.1 | 17.0 | 0.1 | 193.0 | 1.6 |
| 13B | 2.9 | 0.2 | 42.3 | 3.5 | 3.1 | 0.1 | 32.0 | 1.5 | 23.0 | 0.2 | 261.0 | 2.2 |
| 14B | 1.2 | 0.1 | 43.4 | 3.6 | 1.6 | 0.1 | 32.0 | 1.5 | 19.0 | 0.2 | 287.0 | 2.4 |
| 15B | 0.8 | 0.1 | 40.6 | 3.4 | 1.5 | 0.1 | 35.0 | 1.6 | 15.0 | 0.1 | 280.0 | 2.3 |
| Average bituminous coal | 2.2 | 0.2 | 48.9 | 4.1 | 2.0 | 0.1 | 35.4 | 1.6 | 28.0 | 0.2 | 508.3 | 4.2 |

Higher Ga contents were observed in the studied ashes. In the case of lignite ashes, the average content was 27.2 ppm Ga, which is below the Clarke value for coal. However, Ga contents above 90 ppm were observed locally, e.g., in xylitic coal from the Bełchatów deposit (Figure 5). Bituminous coal ashes contained an average of 50.9 ppm Ga. This value was higher than the Clarke number. The highest Ga contents were recorded in bright coal ashes from the Bogdanka deposit; the Ga content in the coal ash from the No. 385/2 seam was 110.4 ppm.

Based on the correlation analysis, it has been found that the content of Sc and V increases with the increase in the Ga content in coal (Figure 6) (Appendix A). Similar relationships were observed for the Ga content in ash. In addition, in the case of Ga content, its content in the ash strongly correlated with the content of $Al_2O_3$ in ash (Figure 7). Similar results were obtained by [78]. This indicates that Ga is largely associated with mineral matter, mainly aluminosilicates. This phenomenon shows that although Ga is found mostly in inorganic carbonaceous matter, not all Ga replace Al in clay minerals in the form of isomorph. According to the abovementioned correlation analysis, it appears that Ga in coal in the form of isomorph tends to remain in the ash after the combustion of coal, while other occurrences of Ga (organic phase) are easily volatile [78]. It has also been found that the Ga content in ash is positively correlated with the $P_2O_5$ content in ash while negatively correlated with the MnO content. At the same time, moderate negative correlations between the Ga content in ash and the total porosity of coal and volatile matter content were observed. Meanwhile, positive correlations with

carbon content in coal and aromaticity factor (fa) were recorded. Based on these results, it should be assumed that the Ga content is higher in coal with a higher degree of coalification. On the other hand, a negative correlation between the Ga content in ash and the carbon content (TOT/C) was also revealed, which indicates that Ga has a strict affinity for the inorganic fraction in coal. Similar results were obtained by Dill and Wehner [83], who found that Ga is associated with layered carbonate silicates; their research carried out in the Schirnding Basin in Germany confirmed negative correlations between Ga and total organic carbon (TOC). Further studies also confirmed these results [79]. It has also been found that sulfide minerals, such as sphalerite, can also contain Ga [84], and its content is strongly influenced by magmatic hydrothermal fluids [85,86]. In the Bogdanka deposit, where high Ga contents are observed, a rich sulfide mineralization is also present [87]. Based on the research, it was found that this mineralization is epigenetic to coal. Other bituminous coal samples show similar mineralization with pyrite and marcasite. This is most likely related to hydrothermal activity, but this issue requires further research. The petrographic analysis of bituminous coal confirmed the occurrence of clay minerals, quartz, dolomite, and sulfides such as pyrite, marcasite, and sphalerite. Quartz, clay minerals and pyrite are visible in lignite. However, due to the fact that the minerals found in the coal samples are very often small or impregnate macerals, it is very difficult to determine their actual share petrographically. Correlations between individual mineral groups, such as clay minerals, sulfides, and carbonates, and critical elements are not statistically significant. Therefore, it is suggested that the results of chemical analysis, which show the content of individual components throughout the sample, are the best material for statistical analysis.

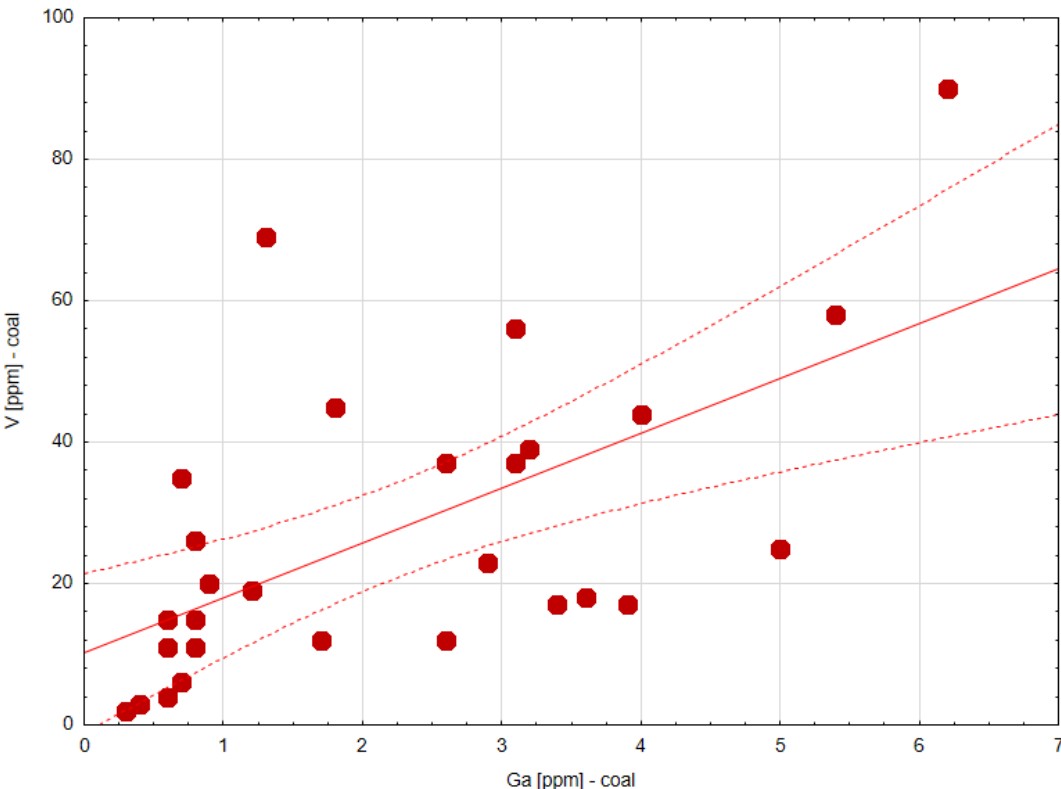

**Figure 6.** Correlation between Ga and V content in coal.

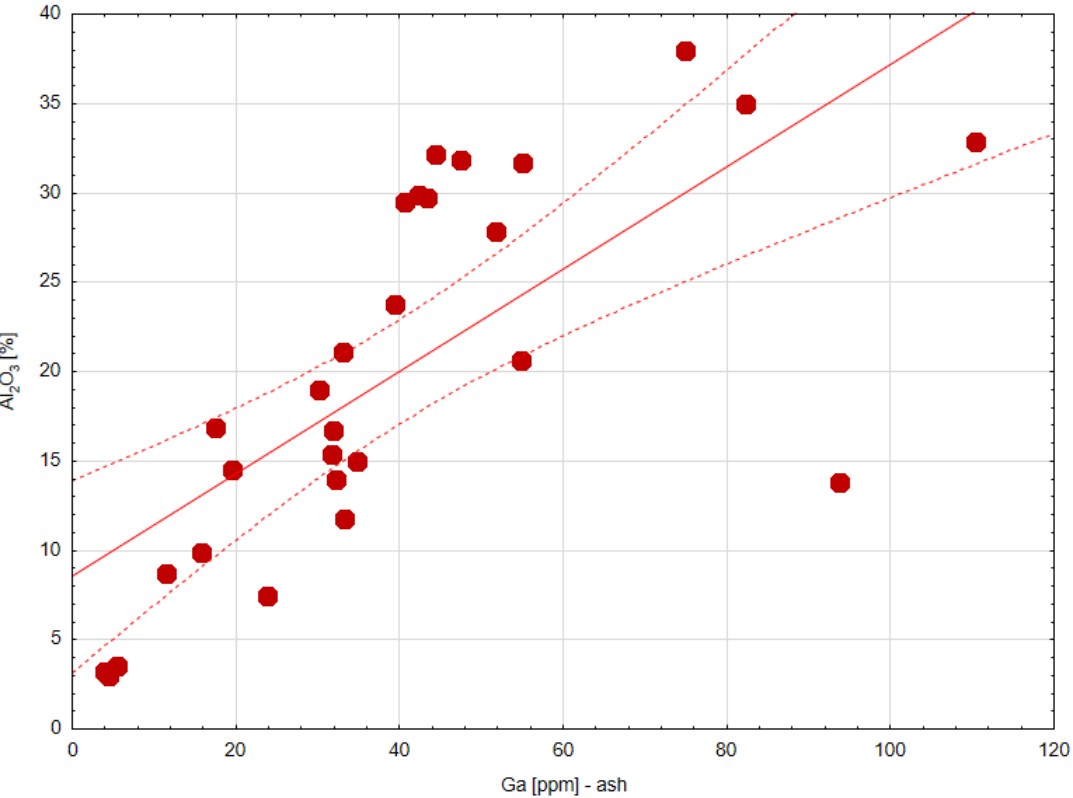

**Figure 7.** Relationship between Ga and $Al_2O_3$ content in coal ash.

### 3.4.2. V—Vanadium

Vanadium belongs to the second group of critical elements. Vanadium is commonly used in a variety of areas, starting from high-strength steels, through to Ti–V–Al alloys, REDOX vanadium batteries, catalysts, specialized ceramics, low-density and high-intensity magnets, cermets, and ending with vanadium redox flow batteries, a competitive alternative to lithium ion batteries, as well as oxidation catalysts in industrial organic chemistry and construction materials for nuclear reactors. The annual production of vanadium is approximately 100,000 tons [88]. The largest producer is China (35% of world production), followed by Russia (25%) and South Africa (20%).

The Clarke value of vanadium in the Earth's crust is 120 ppm [89]. The vanadium minerals are usually dispersed. Vanadium is co-occurring with Fe, Ti, Pb, Zn, and Cu. Usually, vanadium does not form its own independent deposits. In China, about 87% of vanadium is obtained from stone coal [90], i.e., a combustible, low-heat value, high-rank black shale [91]. The Clarke value of V in lignite is 22 ppm, while in bituminous coal it is 28 ppm (25 ppm on average). In coal ash, it is 140 ppm for lignite and 170 ppm for bituminous coal (an average of 155 ppm) [68], respectively. In the case of Polish coal deposits, the average Clarke value of V is 27 ppm (with the standard deviation of 27 ppm). A maximum of 90 ppm was recorded in the Bełchatów region (xylitic coal), while when it comes to bituminous coal from the Bogdanka deposit (Figure 8), the maximum value was 56 ppm. It should be noted that the highest vanadium content was observed in lignite deposits formed in deep grabens, such as Bełchatów, Szczerców, and Turów. While no significant enrichment in vanadium (compared to the Clarke value in coal) was observed, maximum values of 1426 ppm in ash from xylitic lignite from the Turów deposit and 1277 ppm in ash from bituminous coal from the Bogdanka region are reported. The average vanadium content for the tested samples was 458.2 ppm, with a high standard deviation of 429.1 ppm. Therefore, it can be stated that there was a great variation in the vanadium content in the tested samples.

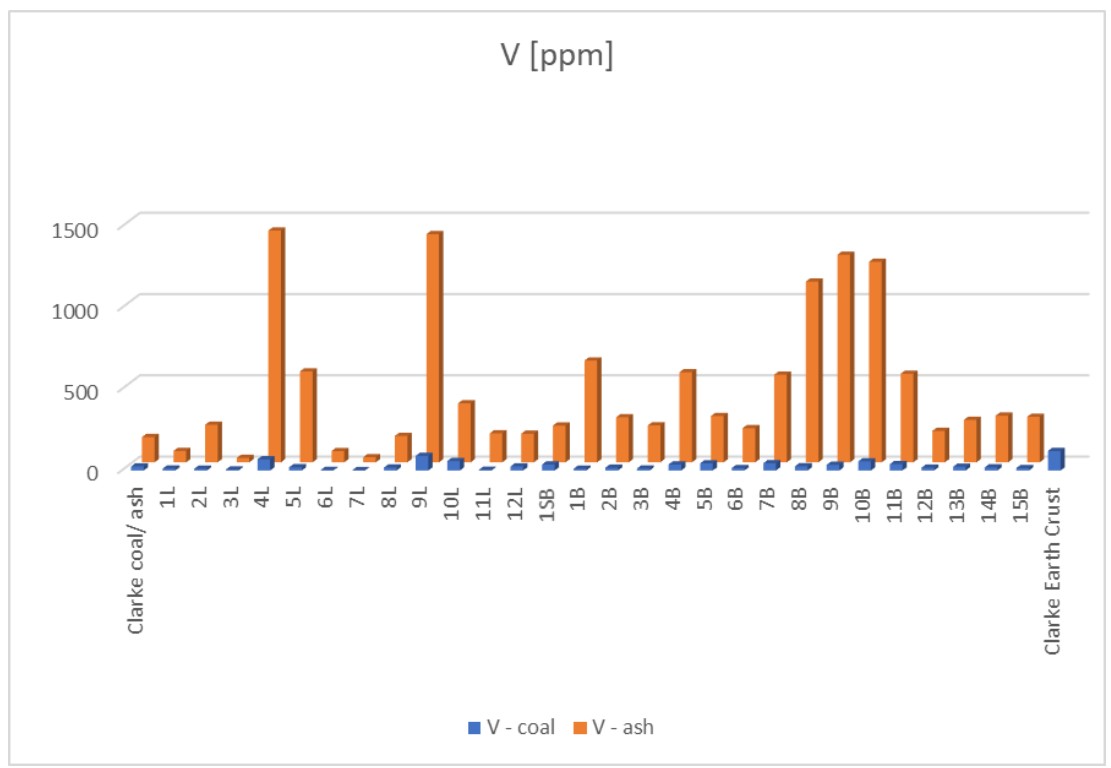

**Figure 8.** Vanadium content in coal and ash in Polish deposits.

The correlation analysis confirmed that the vanadium content in coal increases with the increase in Ga (Figure 7), Sc and total sulfur content ($S_t^{db}$). Similar relationships were also observed for vanadium content in ash (Figure 9). Interestingly, a negative correlation was also found between vanadium in ash and the content of $Al_2O_3$ in ash, ash in coal, and the mineral matter and clay minerals content (Appendix A). Such a correlation suggests that vanadium is associated with organic matter. Similar results were obtained by [92], who confirmed that the $VO^{2+}$ ion is in an environment with an approximate axial symmetry and possibly chelated by carboxylic/phenolic oxygen ligand donor atoms. Such an enrichment took place at the stage of peat formation as a result of vanadium deposited in a swamp environment. It is suggested that the source of vanadium is volcanic ash. The remains of this ash are found in coal deposits in the form of tonsteins. In the case of the examined coal seams, significant amounts of vanadium were accompanied by tonsteins in bituminous coal [93,94] and paratonsteins in lignite [95].

### 3.4.3. Sc—Scandium

Scandium has historically been classified as a rare earth element and is regarded as a critical element. It is mainly used as an aluminum alloy. The $Al_{20}Li_{20}Mg_{10}Sc_{20}Ti_{30}$ alloy is as strong as titanium, as light as aluminum, and as hard as ceramics. These alloys are used as components in the aerospace industry and in mercury lamps. Scandium is present in the Earth's crust at a concentration of about 22 ppm [89], however, it is very dispersed. The only mineral with a high content of this element is thortveitite ($Sc_2Si_2O_7$), found in Norway. Natural scandium deposits are found in Australia, China, Kazakhstan, Russia, Ukraine, the USA, and Madagascar [96]. Due to the rare occurrence of thortveitite, it is not an important source for obtaining scandium on an industrial scale. It is mainly obtained as a by-product during tungsten production and the processing of uranium ore. The global supply and consumption of scandium are around 15 to 20 tons per year. Recently, it was produced as a byproduct material in China (iron ore, rare earths, titanium, and zirconium), Kazakhstan (uranium), Philippines (nickel), Russia (apatite and uranium), and Ukraine (uranium) [97].

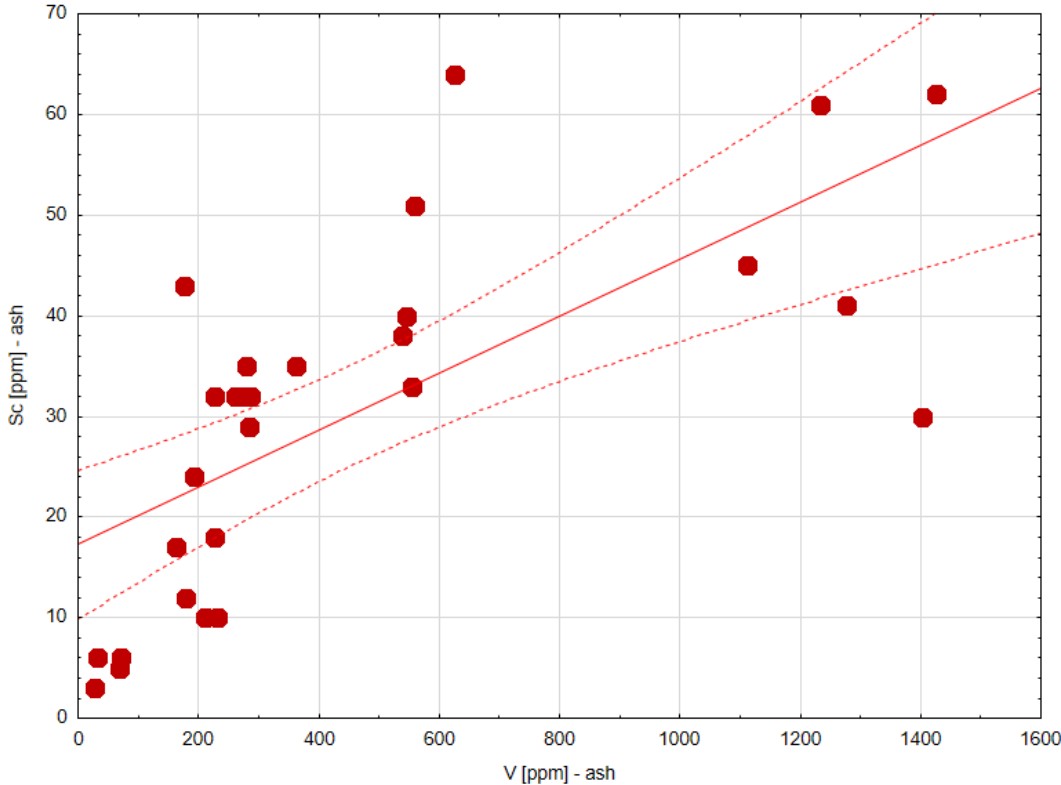

**Figure 9.** Relationship between V and Sc content in coal ash.

The Clarke value of scandium in coal is 3.9 ppm (4.1 ppm for lignite and 3.7 ppm for bituminous coal) [68]. In coal ashes, it amounts to 24 ppm. As of now, there is no large-scale recovery of scandium from coal or its ashes. However, operations at the Jeddo Basin near Hazelton, PA, USA, suggest the possibility of recovering scandium from coal by leaching [98]. Arbuzov et al. [99] have shown that in some deposits of Asian Russia, Mongolia, and Kazakhstan, the scandium content reaches 230 ppm (Pereyaslovka deposit—Kansk–Achinsk basin). In Chinese deposits, the scandium content is 18.3 ppm [100]. In the U.S. deposits, the average scandium content is 4.2 ppm (with a maximum of 100 ppm) [101].

The average scandium content in the tested samples of Polish coal is 2.3 ppm in lignite and 1.9 ppm in bituminous coal. The maximum content is 8 ppm in xylo-detritic lignite from the Szczerców deposit (Figure 10). Generally, the scandium content in Polish coal is lower than the Clarke value for coal. In ashes, the scandium content is higher and amounts to 23.3 ppm for lignite and 36.5 ppm in bituminous coal. The average ash content in Polish deposits is 30.2 ppm, which is higher than the Clarke value for coal ashes. The scandium content in ash above the Clarke value is observed in the Turów and Bełchatów deposits and in virtually all examined bituminous coal deposits (Figure 10).

The correlation analysis has confirmed that the scandium content in coal and ash is moderately positively correlated with the content of gallium (Figure 11) and vanadium (Figure 9). At the same time, a high correlation between the content of $Al_2O_3$ and scandium was revealed (Figure 12). In addition, moderate positive correlations between the scandium content in ash and the content of $Na_2O$, $K_2O$, and $TiO_2$ were confirmed. On the other hand, the scandium content decreases with the increase in CaO and MnO content in the ash. There is also a clear relationship between the rank of coal expressed in gross calorific value ($GCV^{daf}$), carbon content ($C^{daf}$) and total porosity and the scandium content in the ash. As mentioned earlier, higher content of this element was found in coals of higher rank, while the highest content of 64 ppm can be found in coking coal ash.

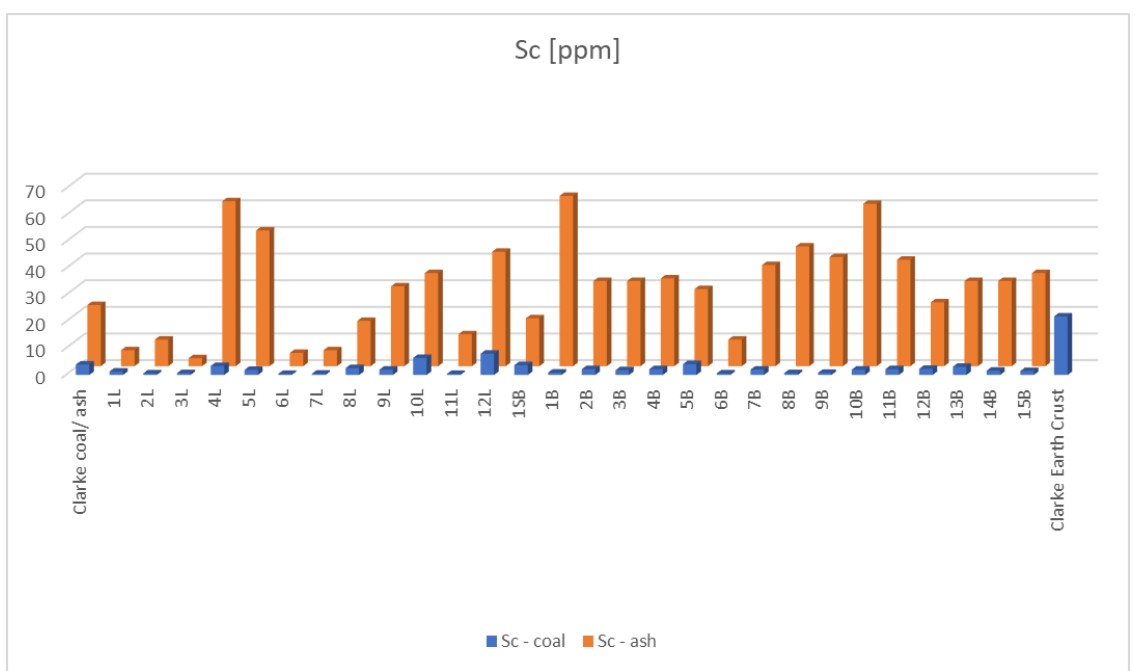

**Figure 10.** The scandium content in coal and ashes from Polish deposits.

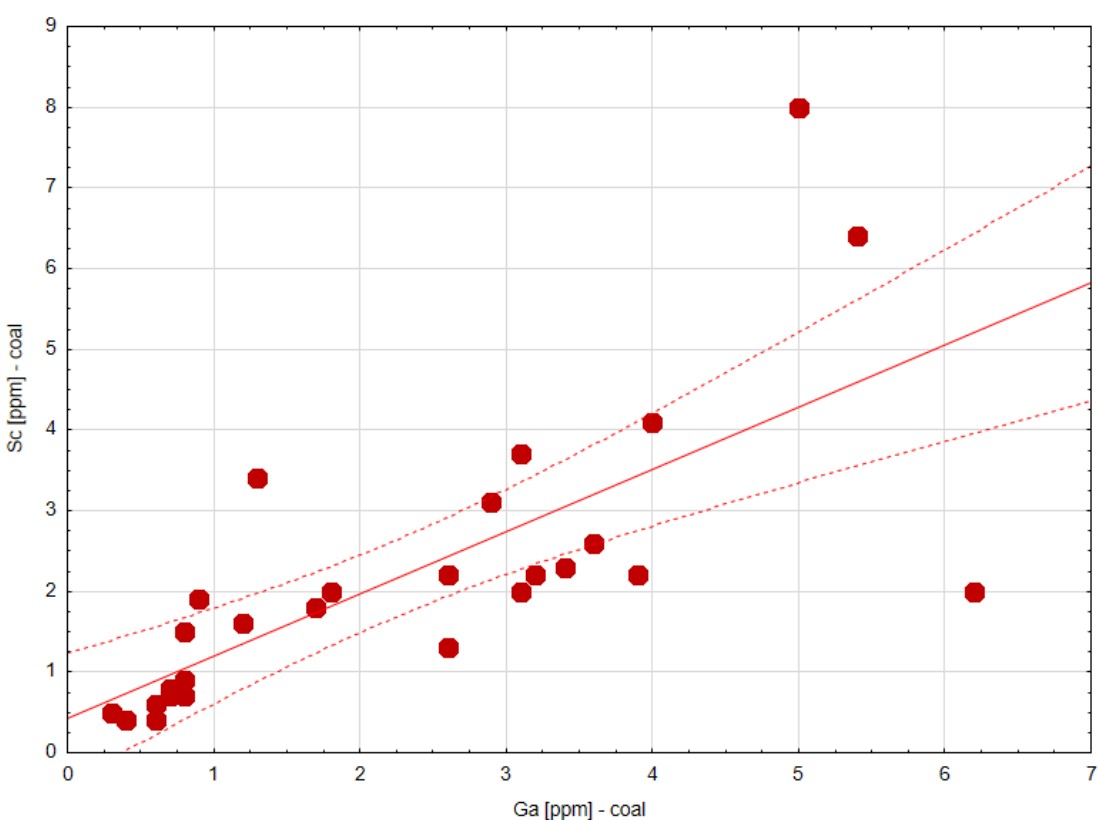

**Figure 11.** Relationship between Ga and Sc content in coal.

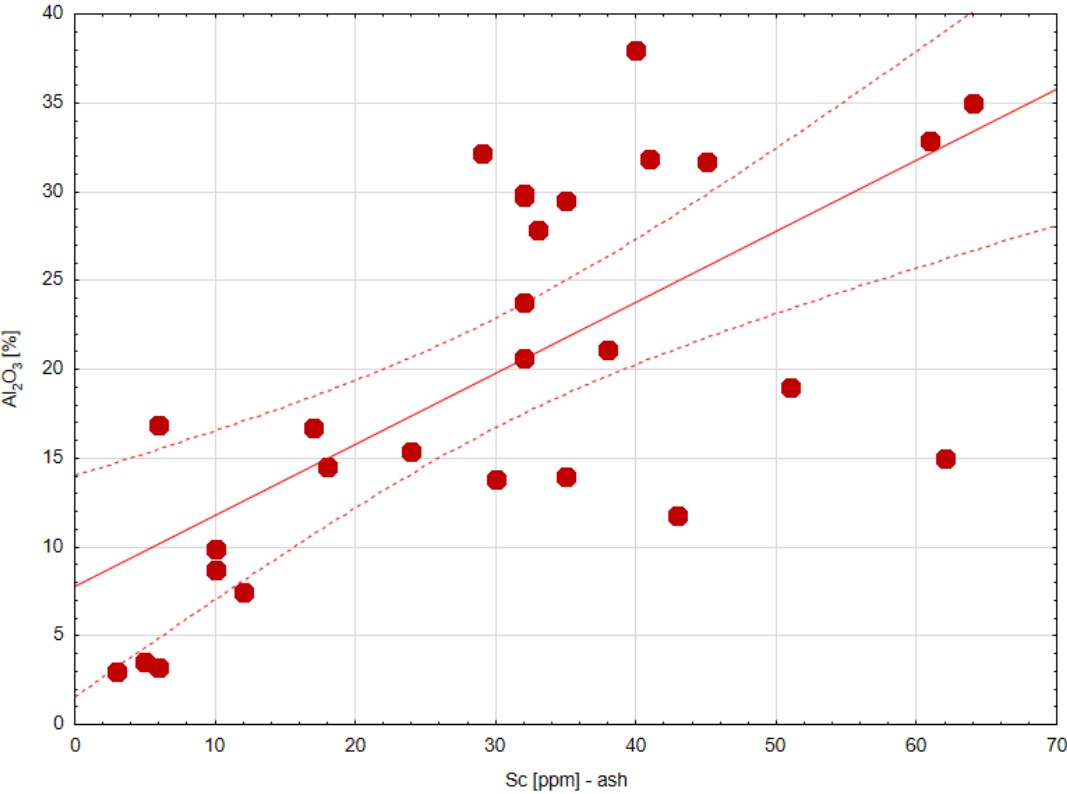

**Figure 12.** Relationship between Sc and $Al_2O_3$ content in coal ash.

The results presented are partly consistent with those presented by Arbuzov et al. [102], who, based on his results, stated that the scandium content in coal decreases with increasing rank of coal due to redeposition. In the case of lignite, where the average scandium content is higher, it is associated with humic acids. The content of humic acids decreases with increasing rank of coal; the scandium is bound to mineral matter, which is manifested by a significant increase in the scandium content in bituminous coal ashes. Such a relationship is also observed in the case of lignite, where higher scandium contents are typical for ashes of coals of higher rank from the Turów and Bełchatów deposits. Based on the correlations carried out, it can be stated that scandium is mainly associated with aluminosilicates.

In order to assess the Ga, Sc, and V content in the tested coals and ashes, their enrichment factors (EFs) were determined. This factor tells how many times the content of a given element is greater than its Clarke value in the Earth's crust. The calculated EFs for individual samples are given in Table 5. The analysis showed that the tested coal is not useful for the recovery of Ga, Sc, and V, unlike the ashes from coal. The EF designated for them is higher compared to the EF in coal. Figure 13 presents EFs for Ga, Sc, and V in the tested ashes. As can be seen, in some cases EF is higher than 10 (e.g., relatively high enrichment factors for vanadium in some of the tested ashes), which is also indicated by color in Table 5. The highest EF for vanadium was determined in the ash from xylitic coal from the Turów deposit (sample 4L) and xylitic coal from the Bełchatów deposit (sample 9L). In both cases, the ash from xylitic coal combustion had the highest vanadium content. It is unclear whether this is related to the petrographic composition of the original sample or epigenetic factors. According to the previous analysis, it can be stated that the vanadium was originally associated with organic matter. The xylitic coal is principally composed of textinite and ulminite, which are often impregnated with liptinite macerals, mostly resinite. It should be noted that this structure favors adsorption of various metals. The abundance of granite in the substrate of the Turów deposit can be associated with solutions rich in V. In addition, the Turów and Bełchatów deposits are cut by numerous faults. It is possible that these tectonic discontinuities were the path of migration of solutions rich in various types of metals. Locally, cracks in the macerals are filled with iron sulfides.

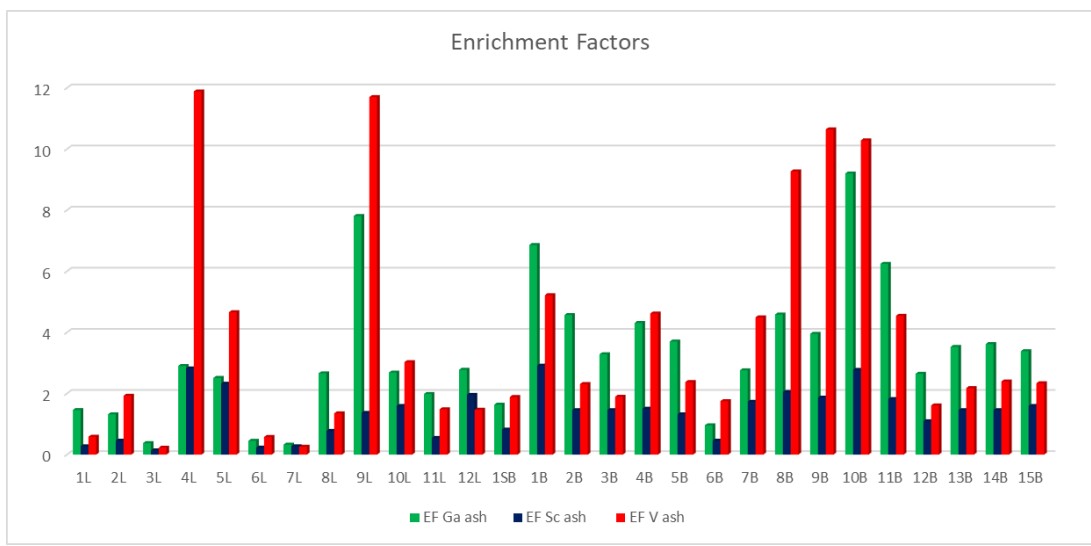

**Figure 13.** Enrichment Factors Ga, Sc, V in coal ash against the Clarke value in the Earth's crust.

In the case of bituminous coal, the greatest enrichments are observed in ashes obtained from coal from the Bogdanka deposit located in the Lublin Coal Basin. The Bogdanka S.A. is the *only coal mine* in the *Lublin Coal Basin (LCB)* that has not yet been explored in detail. The LCB, located in the southwestern part of the Precambrian platform, has been explored since the beginning of the 1960s. The geological structure of the LCB is not complicated; it is located in the East European Platform and tectonic disturbances are not prominently recorded. The highest EFs for V were measured in coal ash from seams No. 381 (sample 9B) and 385/2 (sample 10B). Additionally, increased EFs for Ga in ash were observed in the mentioned seams. Based on earlier studies, it can be concluded that the LCB coal is enriched in critical elements [83]. The exact genesis of such enrichment requires further research.

## 4. Conclusions

The analysis of the coal's properties and its chemical composition confirmed that the ash derived from coal is a key element determining the possibility of using this raw material in a particular technology. Based on the correlation analysis, a number of relationships between the chemical composition of ash and proximate and ultimate analyses, which, already at the initial stage of deposit analysis, indicate its possible applications in various technologies, were demonstrated. In particular, the suitability for individual gasification technologies can be predicted. This is due to the fact that the ash oxide composition greatly influences the flow rate and fouling parameters. The chemical composition of ash is highly dependent on sedimentation conditions and the type of mineral matter supplied to the bog where the coal was formed. Taking into account the ash parameters, the best properties are shown by coal from the Janina deposit, which is located in the eastern part of the Upper Silesian Coal Basin.

Based on an analysis of the composition of ash and the content of critical elements in Polish coals, it can be stated that ashes can be a potential source of some raw materials. However, it should be borne in mind that it will be necessary to intensively enrich the resulting ashes in order to obtain economically viable concentrates. The examined contents of Ga, V, and Sc in Polish deposits are much lower than those in Chinese deposits, but due to the amount of waste from the combustion of coal, the possibility of using them to obtain these elements should be considered.

The analysis has shown that increased levels of Ga, and Sc in the ash are associated with $Al_2O_3$. Therefore, it is suggested that they are associated with aluminosilicates, mainly clay minerals.

When it comes to V, its genesis in coal and ash is unclear and requires further study. On the one hand, the V content increases with an increase in the amount of $Al_2O_3$, but on the other hand it decreases

with increasing amount of mineral matter in coal. Therefore, it is assumed that vanadium originally bound to the organic matter in the combustion process is absorbed by clay minerals. The increased contents of critical elements are, to a large extent, the result of processes that took place after the deposition of peat. In particular, special emphasis should be placed on the migration of elements, acting as sorbents of Ga, V and Sn, from the organic matter to clay minerals. Based on the determined enrichment factors, it can be stated that only ashes from the Turów, Bełchatów, and Bogdanka deposits are rich in Ga, Sc, and V. Due to the highest concentration of the examined critical elements in coal and ash, the most promising is the region of Lublin Coal Basin. Supra-Clarke contents of Ga, V, and Sc were found in coal from the Bogdanka deposit, the only mine of the Lublin Coal Basin. Taking into account the plans to open new mines in the Lublin Coal Basin, the amount of ashes with high concentrations of critical elements is expected to increase. Therefore, basic studies on the content of critical elements, aimed at estimating their potential resource base, are of great importance. Furthermore, a method to enrich the ashes obtained in order to be able to recover Ga and V, e.g., by hydrometallurgical methods, shall be developed.

**Funding:** This research was funded by the Polish National Science Centre under research project awarded by decision No. DEC-2013/09/D/ST10/04045 and from subsidy No. 16.16.140.315.

**Conflicts of Interest:** The author declares no conflict of interest.

## Appendix A

**Table A1.** Correlation between ash oxide composition and proximate, ultimate, and petrographic analyses.

| | $SiO_2$ (%) | $Al_2O_3$ (%) | $Fe_2O_3$ (%) | MgO (%) | CaO (%) | $Na_2O$ (%) |
|---|---|---|---|---|---|---|
| $M^{ad}$ (%) | −0.3 | −0.7 | −0.3 | 0.1 | 0.7 | 0.1 |
| $A^{db}$ (%) | 0.6 | −0.1 | −0.3 | −0.3 | −0.1 | −0.3 |
| $V^{daf}$ (%) | −0.2 | −0.8 | −0.2 | 0.1 | 0.7 | 0 |
| $S_t^{db}$ (%) | 0 | −0.3 | 0.5 | −0.4 | 0 | −0.2 |
| GCV (MJ/kg) | 0.1 | 0.8 | 0.3 | 0 | −0.7 | 0.1 |
| Ash Sintering Temperature (°C) | −0.1 | −0.2 | 0.1 | 0.1 | 0 | 0.3 |
| Ash Softening Temperature (°C) | 0.5 | 0.4 | 0 | −0.5 | −0.5 | −0.3 |
| Ash Melting Temperature (°C) | 0.5 | 0.4 | 0.2 | −0.4 | −0.6 | −0.3 |
| Ash Fluid Temperature (°C) | 0.5 | 0.4 | 0.2 | −0.4 | −0.7 | −0.2 |
| $C^{daf}$ (%) | 0.1 | 0.8 | 0.3 | −0.1 | −0.7 | 0.1 |
| $H^{daf}$ (%) | −0.3 | −0.2 | 0.1 | 0.3 | 0 | 0.4 |
| $N^{daf}$ (%) | 0 | 0.6 | 0.5 | −0.2 | −0.6 | −0.1 |
| $O^{daf}$ (%) | −0.1 | −0.8 | −0.4 | 0.1 | 0.7 | −0.1 |
| Total porosity (%) | 0.1 | −0.7 | −0.3 | 0.1 | 0.5 | 0 |
| Huminite/Vitrinite (%) | −0.3 | −0.7 | −0.1 | 0.2 | 0.5 | 0 |
| Inertinite (%) | 0.3 | 0.7 | −0.1 | −0.2 | −0.5 | 0 |
| Liptinite (%) | 0.1 | 0.6 | 0.2 | −0.2 | −0.5 | 0 |
| Mineral Matter (%) | 0.2 | −0.4 | 0 | −0.2 | 0.1 | −0.3 |
| Random Reflectance (Ro) (%) | 0.1 | 0.8 | 0.2 | −0.1 | −0.6 | 0 |
| Aromaticity Factor (fa) | 0.3 | 0.7 | 0.2 | −0.2 | −0.6 | −0.1 |
| $SiO_2$ (%) | 1 | 0.3 | −0.4 | −0.4 | −0.6 | −0.2 |
| $Al_2O_3$ | 0.3 | 1 | −0.1 | −0.2 | −0.7 | 0.1 |
| $Fe_2O_3$ (%) | −0.4 | −0.1 | 1 | −0.1 | −0.3 | 0 |
| MgO (%) | −0.4 | −0.2 | −0.1 | 1 | 0.1 | 0.7 |
| CaO (%) | −0.6 | −0.7 | −0.3 | 0.1 | 1 | −0.3 |
| $Na_2O$ (%) | −0.2 | 0.1 | 0 | 0.7 | −0.3 | 1 |
| $K_2O$ (%) | 0.3 | 0.6 | −0.1 | 0.1 | −0.6 | 0.3 |
| $TiO_2$ (%) | 0.5 | 0.4 | −0.3 | 0.2 | −0.6 | 0.3 |
| $P_2O_5$ (%) | 0 | 0.5 | 0 | 0.1 | −0.2 | −0.1 |
| MnO (%) | −0.4 | −0.4 | 0.5 | 0.2 | 0.2 | −0.2 |
| $Cr_2O_3$ (%) | 0.5 | −0.2 | −0.1 | 0 | −0.2 | 0 |
| LOI and Others (%) | −0.7 | −0.8 | −0.2 | 0.3 | 0.9 | 0 |

**Table A1.** *Cont.*

| | | | | | | |
|---|---|---|---|---|---|---|
| C in ash (%) | −0.5 | −0.4 | −0.1 | 0.2 | 0.6 | −0.1 |
| S in ash (%) | −0.5 | −0.7 | −0.2 | 0.2 | 0.9 | −0.1 |
| Base/Acid (B/A) ratio | −0.7 | −0.6 | 0.5 | 0.1 | 0.5 | −0.1 |
| Slagging Index (Rs) | −0.5 | −0.2 | 0.5 | 0.3 | 0.1 | −0.2 |
| Silica Value (SV) | 0.9 | 0.6 | −0.5 | −0.3 | −0.6 | −0.1 |
| Fouling Index (Rf) | −0.4 | 0 | 0.2 | 0.8 | −0.2 | 0.9 |
| Alkalinity (AK) | −0.2 | 0.1 | 0 | 0.7 | −0.3 | 1 |
| Ga (ppm)—coal | 0.1 | 0.1 | −0.2 | −0.3 | 0.1 | −0.1 |
| Ga (ppm)—ash | 0.1 | 0.7 | −0.1 | −0.2 | −0.3 | 0 |
| Sc (ppm)—coal | 0.2 | 0 | −0.1 | −0.1 | −0.1 | 0 |
| Sc (ppm)—ash | 0.1 | 0.7 | 0.1 | 0.1 | −0.6 | 0.4 |
| V (ppm)—coal | −0.1 | 0.2 | 0.1 | −0.1 | −0.1 | 0.1 |
| V (ppm)—ash | −0.1 | 0.4 | 0.1 | 0 | −0.3 | 0.2 |

| | $K_2O$ (%) | $TiO_2$ (%) | $P_2O_5$ (%) | MnO (%) | $Cr_2O_3$ (%) | LOI and Others (%) |
|---|---|---|---|---|---|---|
| $M^{ad}$ (%) | −0.4 | −0.2 | −0.7 | −0.1 | 0.2 | 0.8 |
| $A^{db}$ (%) | −0.1 | 0.2 | −0.2 | −0.1 | 0.4 | −0.2 |
| $V^{daf}$ (%) | −0.5 | −0.2 | −0.5 | 0 | 0.2 | 0.8 |
| $S_t^{db}$ (%) | −0.3 | −0.1 | −0.3 | 0.1 | 0.1 | 0 |
| GCV (MJ/kg) | 0.4 | 0.2 | 0.5 | 0 | −0.2 | −0.7 |
| Ash Sintering Temperature (°C) | 0 | 0 | −0.4 | −0.1 | 0 | 0.1 |
| Ash Softening Temperature (°C) | 0.2 | 0.2 | 0 | −0.1 | 0.1 | −0.5 |
| Ash Melting Temperature (°C) | 0.2 | 0.2 | 0.1 | 0 | 0.1 | −0.6 |
| Ash Fluid Temperature (°C) | 0.3 | 0.2 | 0.2 | 0.1 | 0.2 | −0.7 |
| $C^{daf}$ (%) | 0.4 | 0.1 | 0.5 | 0 | −0.2 | −0.7 |
| $H^{daf}$ (%) | 0 | 0.2 | −0.2 | −0.2 | 0 | 0.2 |
| $N^{daf}$ (%) | 0.4 | 0.1 | 0.6 | 0.2 | −0.3 | −0.7 |
| $O^{daf}$ (%) | −0.4 | −0.2 | −0.5 | 0 | 0.2 | 0.8 |
| Total porosity (%) | −0.4 | 0 | −0.6 | −0.1 | 0.4 | 0.5 |
| Huminite/Vitrinite (%) | −0.4 | −0.2 | −0.5 | 0.2 | 0.3 | 0.6 |
| Inertinite (%) | 0.4 | 0.1 | 0.4 | −0.3 | −0.2 | −0.6 |
| Liptinite (%) | 0.3 | 0.1 | 0.5 | −0.2 | −0.3 | −0.5 |
| Mineral Matter (%) | −0.1 | 0.1 | −0.3 | 0.1 | −0.1 | 0.1 |
| Random Reflectance (Ro) (%) | 0.3 | 0.1 | 0.5 | 0 | −0.2 | −0.6 |
| Aromaticity Factor (fa) | 0.5 | 0.2 | 0.5 | 0 | −0.2 | −0.8 |
| $SiO_2$ (%) | 0.3 | 0.5 | 0 | −0.4 | 0.5 | −0.7 |
| $Al_2O_3$ | 0.6 | 0.4 | 0.5 | −0.4 | −0.2 | −0.8 |
| $Fe_2O_3$ (%) | −0.1 | −0.3 | 0 | 0.5 | −0.1 | −0.2 |
| MgO (%) | 0.1 | 0.2 | 0.1 | 0.2 | 0 | 0.3 |
| CaO (%) | −0.6 | −0.6 | −0.2 | 0.2 | −0.2 | 0.9 |
| $Na_2O$ (%) | 0.3 | 0.3 | −0.1 | −0.2 | 0 | 0 |
| $K_2O$ (%) | 1 | 0.4 | 0.1 | −0.3 | −0.1 | −0.5 |
| $TiO_2$ (%) | 0.4 | 1 | 0.1 | −0.5 | 0 | −0.5 |
| $P_2O_5$ (%) | 0.1 | 0.1 | 1 | 0.1 | −0.2 | −0.4 |
| MnO (%) | −0.3 | −0.5 | 0.1 | 1 | 0 | 0.2 |
| $Cr_2O_3$ (%) | −0.1 | 0 | −0.2 | 0 | 1 | −0.2 |
| LOI and Others (%) | −0.5 | −0.5 | −0.4 | 0.2 | −0.2 | 1 |
| C in ash (%) | −0.3 | −0.3 | −0.1 | 0.3 | −0.4 | 0.6 |
| S in ash (%) | −0.5 | −0.4 | −0.5 | 0 | −0.1 | 0.9 |
| Base/Acid (B/A) ratio | −0.4 | −0.5 | −0.2 | 0.6 | −0.3 | 0.6 |
| Slagging Index (Rs) | −0.2 | −0.3 | 0.4 | 0.7 | −0.2 | 0.1 |
| Silica Value (SV) | 0.5 | 0.6 | 0.2 | −0.5 | 0.3 | −0.7 |
| Fouling Index (Rf) | 0.2 | 0.3 | −0.2 | −0.1 | 0 | 0.1 |
| Alkalinity (AK) | 0.3 | 0.4 | −0.1 | −0.2 | 0 | −0.1 |
| Ga (ppm)—coal | 0 | 0.2 | 0 | −0.2 | −0.2 | 0 |
| Ga (ppm)—ash | 0.2 | 0.3 | 0.4 | −0.4 | −0.1 | −0.4 |
| Sc (ppm)—coal | 0.1 | 0.5 | −0.1 | −0.3 | −0.3 | −0.1 |
| Sc (ppm)—ash | 0.4 | 0.6 | 0.3 | −0.5 | −0.2 | −0.5 |
| V (ppm)—coal | 0.2 | 0.2 | 0.1 | −0.3 | −0.1 | −0.1 |
| V (ppm)—ash | 0.3 | 0.3 | 0.2 | −0.3 | 0 | −0.2 |

**Table A1.** *Cont.*

| | C in ash (%) | S in ash (%) | Base/Acid (B/A) Ratio | Slagging Index (Rs) | Silica Value (SV) | Fouling Index (Rf) | Alkalinity (AK) |
|---|---|---|---|---|---|---|---|
| $M^{ad}$ (%) | 0.3 | 0.9 | 0.3 | −0.2 | −0.4 | 0.2 | 0.1 |
| $A^{db}$ (%) | −0.3 | −0.1 | −0.3 | −0.2 | 0.5 | −0.3 | −0.3 |
| $V^{daf}$ (%) | 0.3 | 0.8 | 0.3 | −0.1 | −0.4 | 0.1 | 0 |
| $S_t^{db}$ (%) | −0.4 | 0.1 | 0 | 0.2 | −0.2 | −0.1 | −0.2 |
| GCV (MJ/kg) | −0.2 | −0.7 | −0.2 | 0.1 | 0.2 | 0 | 0.1 |
| Ash Sintering Temperature (°C) | 0.1 | 0.1 | 0.2 | −0.3 | −0.1 | 0.4 | 0.3 |
| Ash Softening Temperature (°C) | −0.4 | −0.5 | −0.3 | −0.1 | 0.6 | −0.4 | −0.3 |
| Ash Melting Temperature (°C) | −0.4 | −0.6 | −0.2 | 0.1 | 0.5 | −0.3 | −0.3 |
| Ash Fluid Temperature (°C) | −0.5 | −0.7 | −0.3 | 0.1 | 0.5 | −0.2 | −0.2 |
| $C^{daf}$ (%) | −0.2 | −0.8 | −0.2 | 0.1 | 0.3 | 0 | 0.1 |
| $H^{daf}$ (%) | 0.1 | 0.2 | 0.2 | −0.2 | −0.2 | 0.5 | 0.3 |
| $N^{daf}$ (%) | −0.2 | −0.7 | 0 | 0.4 | 0.1 | −0.2 | −0.1 |
| $O^{daf}$ (%) | 0.3 | 0.8 | 0.2 | −0.2 | −0.3 | 0 | −0.1 |
| Total porosity (%) | 0.3 | 0.6 | 0.2 | −0.3 | −0.1 | 0.1 | 0 |
| Huminite/ Vitrinite (%) | 0.3 | 0.6 | 0.4 | 0.1 | −0.4 | 0.2 | 0 |
| Inertinite (%) | −0.3 | −0.6 | −0.4 | −0.2 | 0.5 | −0.2 | 0 |
| Liptinite (%) | −0.2 | −0.6 | −0.2 | 0.1 | 0.2 | −0.1 | 0 |
| Mineral Matter (%) | 0.3 | 0 | 0.3 | −0.1 | 0 | −0.2 | −0.3 |
| Random Reflectance (Ro) (%) | −0.2 | −0.7 | −0.2 | 0.2 | 0.3 | −0.2 | 0 |
| Aromaticity Factor (fa) | −0.3 | −0.8 | −0.3 | 0.2 | 0.4 | −0.2 | −0.1 |
| $SiO_2$ (%) | −0.5 | −0.5 | −0.7 | −0.5 | 0.9 | −0.4 | −0.2 |
| $Al_2O_3$ | −0.4 | −0.7 | −0.6 | −0.2 | 0.6 | 0 | 0.1 |
| $Fe_2O_3$ (%) | −0.1 | −0.2 | 0.5 | 0.5 | −0.5 | 0.2 | 0 |
| MgO (%) | 0.2 | 0.2 | 0.1 | 0.3 | −0.3 | 0.8 | 0.7 |
| CaO (%) | 0.6 | 0.9 | 0.5 | 0.1 | −0.6 | −0.2 | −0.3 |
| $Na_2O$ (%) | −0.1 | −0.1 | −0.1 | −0.2 | −0.1 | 0.9 | 1 |
| $K_2O$ (%) | −0.3 | −0.5 | −0.4 | −0.2 | 0.5 | 0.2 | 0.3 |
| $TiO_2$ (%) | −0.3 | −0.4 | −0.5 | −0.3 | 0.6 | 0.3 | 0.4 |
| $P_2O_5$ (%) | −0.1 | −0.5 | −0.2 | 0.4 | 0.2 | −0.2 | −0.1 |
| MnO (%) | 0.3 | 0 | 0.6 | 0.7 | −0.5 | −0.1 | −0.2 |
| $Cr_2O_3$ (%) | −0.4 | −0.1 | −0.3 | −0.2 | 0.3 | 0 | 0 |
| LOI and Others (%) | 0.6 | 0.9 | 0.6 | 0.1 | −0.7 | 0.1 | −0.1 |
| C in ash (%) | 1 | 0.3 | 0.7 | 0.2 | −0.5 | 0 | −0.1 |
| S in ash (%) | 0.3 | 1 | 0.3 | 0 | −0.6 | 0.1 | −0.1 |
| Base/Acid (B/A) ratio | 0.7 | 0.3 | 1 | 0.4 | −0.8 | 0.1 | −0.1 |
| Slagging Index (Rs) | 0.2 | 0 | 0.4 | 1 | −0.5 | −0.1 | −0.2 |
| Silica Value (SV) | −0.5 | −0.6 | −0.8 | −0.5 | 1 | −0.3 | −0.1 |
| Fouling Index (Rf) | 0 | 0.1 | 0.1 | −0.1 | −0.3 | 1 | 0.9 |
| Alkalinity (AK) | −0.1 | −0.1 | −0.1 | −0.2 | −0.1 | 0.9 | 1 |
| Ga (ppm)—coal | −0.3 | 0.1 | −0.3 | 0.1 | 0.1 | −0.2 | −0.1 |
| Ga (ppm)—ash | −0.4 | −0.3 | −0.4 | 0 | 0.3 | −0.1 | 0 |

**Table A1.** *Cont.*

| | | | | | | |
|---|---|---|---|---|---|---|
| Sc (ppm)—coal | −0.3 | 0 | −0.3 | −0.1 | 0.2 | 0 | 0 |
| Sc (ppm)—ash | −0.4 | −0.4 | −0.4 | −0.1 | 0.2 | 0.4 | 0.4 |
| V (ppm)—coal | −0.4 | 0.1 | −0.2 | 0.1 | 0 | 0.1 | 0.1 |
| V (ppm)—ash | −0.3 | −0.1 | −0.2 | 0.1 | 0 | 0.3 | 0.2 |

| | Ga (ppm)—coal | Ga (ppm)—ash | Sc (ppm)—coal | Sc (ppm)—ash | V (ppm)—coal | V (ppm)—ash |
|---|---|---|---|---|---|---|
| $M^{ad}$ (%) | 0.1 | −0.5 | 0.1 | −0.5 | 0 | −0.3 |
| $A^{db}$ (%) | 0.4 | −0.3 | 0.4 | −0.4 | 0 | −0.4 |
| $V^{daf}$ (%) | 0.1 | −0.5 | 0.1 | −0.5 | 0 | −0.2 |
| $S_t^{db}$ (%) | 0.4 | −0.1 | 0.4 | 0 | 0.4 | 0.1 |
| GCV (MJ/kg) | −0.3 | 0.5 | −0.2 | 0.6 | 0 | 0.4 |
| Ash Sintering Temperature (°C) | 0 | −0.2 | 0.2 | 0.1 | 0 | 0 |
| Ash Softening Temperature (°C) | 0.1 | 0.3 | 0.1 | 0.2 | 0.1 | 0.1 |
| Ash Melting Temperature (°C) | 0.1 | 0.4 | 0 | 0.2 | 0.2 | 0.1 |
| Ash Fluid Temperature (°C) | 0.1 | 0.4 | 0 | 0.2 | 0.1 | 0.1 |
| $C^{daf}$ (%) | −0.3 | 0.5 | −0.2 | 0.6 | 0 | 0.3 |
| $H^{daf}$ (%) | −0.4 | −0.1 | −0.2 | 0.2 | 0.1 | 0.3 |
| $N^{daf}$ (%) | −0.1 | 0.3 | 0 | 0.4 | 0.1 | 0.2 |
| $O^{daf}$ (%) | 0.2 | −0.5 | 0.1 | −0.6 | 0 | −0.3 |
| Total porosity (%) | −0.1 | −0.5 | 0 | −0.5 | −0.3 | −0.3 |
| Huminite/Vitrinite (%) | 0 | −0.2 | −0.1 | −0.2 | 0 | 0 |
| Inertinite (%) | −0.1 | 0.3 | 0 | 0.3 | −0.1 | 0 |
| Liptinite (%) | 0 | 0.2 | 0.1 | 0.3 | 0.1 | 0.1 |
| Mineral Matter (%) | 0.2 | −0.4 | 0.5 | −0.3 | −0.1 | −0.4 |
| Random Reflectance (Ro) (%) | −0.1 | 0.5 | −0.2 | 0.5 | 0 | 0.2 |
| Aromaticity Factor (fa) | 0.2 | 0.5 | 0.1 | 0.4 | 0 | 0.1 |
| $SiO_2$ (%) | 0.1 | 0.1 | 0.2 | 0.1 | −0.1 | −0.1 |
| $Al_2O_3$ | 0.1 | 0.7 | 0 | 0.7 | 0.2 | 0.4 |
| $Fe_2O_3$ (%) | −0.2 | −0.1 | −0.1 | 0.1 | 0.1 | 0.1 |
| MgO (%) | −0.3 | −0.2 | −0.1 | 0.1 | −0.1 | 0 |
| CaO (%) | 0.1 | −0.3 | −0.1 | −0.6 | −0.1 | −0.3 |
| $Na_2O$ (%) | −0.1 | 0 | 0 | 0.4 | 0.1 | 0.2 |
| $K_2O$ (%) | 0 | 0.2 | 0.1 | 0.4 | 0.2 | 0.3 |
| $TiO_2$ (%) | 0.2 | 0.3 | 0.5 | 0.6 | 0.2 | 0.3 |
| $P_2O_5$ (%) | 0 | 0.4 | −0.1 | 0.3 | 0.1 | 0.2 |
| MnO (%) | −0.2 | −0.4 | −0.3 | −0.5 | −0.3 | −0.3 |
| $Cr_2O_3$ (%) | −0.2 | −0.1 | −0.3 | −0.2 | −0.1 | 0 |
| LOI and Others (%) | 0 | −0.4 | −0.1 | −0.5 | −0.1 | −0.2 |
| C in ash (%) | −0.3 | −0.4 | −0.3 | −0.4 | −0.4 | −0.3 |
| S in ash (%) | 0.1 | −0.3 | 0 | −0.4 | 0.1 | −0.1 |
| Base/Acid (B/A) ratio | −0.3 | −0.4 | −0.3 | −0.4 | −0.2 | −0.2 |
| Slagging Index (Rs) | 0.1 | 0 | −0.1 | −0.1 | 0.1 | 0.1 |
| Silica Value (SV) | 0.1 | 0.3 | 0.2 | 0.2 | 0 | 0 |
| Fouling Index (Rf) | −0.2 | −0.1 | 0 | 0.4 | 0.1 | 0.3 |
| Alkalinity (AK) | −0.1 | 0 | 0 | 0.4 | 0.1 | 0.2 |
| Ga (ppm)—coal | 1 | 0.4 | 0.7 | 0.1 | 0.6 | 0.1 |
| Ga (ppm)—ash | 0.4 | 1 | 0.1 | 0.7 | 0.6 | 0.7 |
| Sc (ppm)—coal | 0.7 | 0.1 | 1 | 0.3 | 0.4 | 0 |
| Sc (ppm)—ash | 0.1 | 0.7 | 0.3 | 1 | 0.5 | 0.7 |
| V (ppm)—coal | 0.6 | 0.6 | 0.4 | 0.5 | 1 | 0.7 |
| V (ppm)—ash | 0.1 | 0.7 | 0 | 0.7 | 0.7 | 1 |

XX—statistically significant at = 0.05.

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
