# Peer review of "Ash Characteristics and Selected Critical Elements (Ga, Sc, V) in Coal and Ash in Polish Deposits"

_resources, doi:10.3390/resources9090115_

Round 1
Reviewer 1 Report
I reviewed this manuscript previously. I have now re-read it thoroughly, and I believe that the author has made a good-faith effort to address the principal concerns of my previous review. Therefore my recommendation is to proceed with publication.
Author Response
Thank you for your positive review of the manuscript. However, according other reviewers comments, a few changes have been made. The English language was checked by English Editing Publishing Services (MDPI), English editing ID: english-21773.
Reviewer 2 Report
The organization and discussion of MS are good; however, some points need to be re-organize by author. Firstly, if the author is not published the results of standard coal quality and petrographical compositions previousy somewhere, these results should be added as a separate subchapter in results and discussion chapter rather than chapter 2. Furthermore, if it is possible, author should provide some snaps of macerals and mineral matter from coal petrography analysis. The latter one could be useful for mineralogy and mode of occurrence of selected elements. Secondly, the author should provide a table which is reporting the identified minerals in the studied coal and ash samples. Finally, the author should summarize and provide less literature data about Ga, V and Sc data in the related subchapters. I added some suggestions ain the attached revised MS. The author should read and pay attention suggestions as seen post-it in attached pdf. Overall, I would like to re-consider the MS, after suggested corrections done.

Author Response
The authors would like to thank for valuable comments and linguistic corrections.
Some of the specific comments need to be answered:
The organization and discussion of MS are good; however, some points need to be re-organize by author. Firstly, if the author is not published the results of standard coal quality and petrographical compositions previousy somewhere, these results should be added as a separate subchapter in results and discussion chapter rather than chapter 2.
Detailed petrographic data of the tested samples are presented in another article that is currently under review. Therefore, the complete petrographic data is not included in order to avoid self-plagiarism.
Furthermore, if it is possible, author should provide some snaps of macerals and mineral matter from coal petrography analysis. The latter one could be useful for mineralogy and mode of occurrence of selected elements. Secondly, the author should provide a table which is reporting the identified minerals in the studied coal and ash samples.
An additional chapter discussing the general petrographic description of the tested samples was included. Unfortunately, microscopic petrographic analysis was carried out only for coal; because of the fact that the obtained ashes were very fragmented and I had a limited amount of samples, I decided to perform XRD and ICP-MS tests. I believe that there is no need to include the entire XRD analysis here, because the mineralogical composition of the tested coal and ashes is not complicated and is limited to a few minerals that are listed in the article.
Finally, the author should summarize and provide less literature data about Ga, V and Sc data in the related subchapters.
Dear Reviewer! Unfortunately, your suggestions regarding literature data are contradictory to other reviewers' suggestions. Therefore, I decided to follow only some of the comments, as I had to choose between some of the comments as they were conflicting. At the same time, I also believe that it is beneficial to use as much literature data as possible, as they will be of interest to the readers of the journal.
I added some suggestions in the attached revised MS. The author should read and pay attention suggestions as seen post-it in attached pdf.
Line 25
The suggested countries have been added.
Line 32 Even though applications on coal combustion remains are well-know, it is better to provide some references in the introduction chapter
References in the introduction chapter have been added.
Line 52
Done.
Line 76 Please add following sentence:
The mineralogical and elemental compositions of coal combustion remains have been subjected by several studies during last three decades due to techinical and environmental concerns.
Please check and cite following studies:
- Dai et al., 2010. Int J Coal Geol. 81, 320-332.
- Fernandez-Turiel et al., 2004. Energy and Fuels 18, 1512-1518.
- Filippidis et al. 1996. Int J Coal Geol. 30, 303-314.
- Goodarzi, 2006. Int J Coal Geol. 65, 17-25
- Huffman et al., 1990. Progress in Energy and Combustion Science 16, 243-251.
- Karayigit et al., 2019. Turk. J. Earth Sci. 28, 438-456.
- Karayigit et al. 2019. Coal Combustion and Gasification Products 11, 18-31.
- Kostova et al., 2013. Int J Coal Geol. 116-117, 227-235.
- Vassilev et al., 2005. Int J Coal Geol. 61, 35-63.
The sentence has been added. The abovementioned studies have been cited.
Line 120 It would be useful to provide some snaps of macerals and mineral matter from coal petrography studies. The latter one is essential for discussions about mentioned mineral matters in mode of Ga, V and Sc.
A table and a description of the petrographic composition and the share of minerals in the tested samples have been added.
Line 133 Please provide references! Several studies, which you have already cited in your MS, were focused on mineral formatiın during coal combustion
References have been provided.
Line 162 It would be nice to see a table that is holding identifying minerals from studied coal and coal ashes.
I have not conducted a detailed research on the amount of individual minerals. In the XRD analysis, virtually all samples contained similar minerals, which were described in the text. I would like to avoid generating more tables as the article is already quite long at this point.
Line 172 Please check and cite following studies:
- Hower et al., 2017. Int J Coal Geol. 179, 11-27.
- Karayigit et al. 2019. Coal Combustion and Gasification Products 11, 18-31.
- Styszko-Grochowiak et al., 2004. Fuel 83, 1847–1853.
Done.
Line 183 Ca in coal could be affilated with organic matter sulphates (e.g. gypsum) and siderite.
Please check and cite following studies:
- Dai et al., 2007. Sci. China Earth Sci. 50, 678-688
-Karayigit et al., 2017. Int. J Coal Geol. 173,110-128.
- Ward, 2016. Int J Coal Geol., 165, 1-27.
The references have been supplemented.
Line 187 Just a recommendation for your future studies, please combine your results from statistical method and XRD data with SEM-EDX data.
Thank you for your valuable comment.
Line 192 - or high detrital input ratios and/or sedimant-laden water influences into palaeomires.
Please check following studies:
- Christanis et al., 1998. Int. J Coal Geol.36, 295-313
- Kalaitzidis et al., 2002. Energy and Fuels
16, 1476-1482.
- Karayigit et al., 2017. Int. J Coal Geol. 173,110-128.
- Ward, 2016. Int J Coal Geol., 165, 1-27
The sentence has been rewritten.
Line 237 via should be italic
Done.
Line 291 it is better to mention highlighted section in the introduction chapter
The sentence has been rewritten, a similar sentence has been included in the introduction.
Line 291 Summerize highlighted part
A summary at the end of the highlighted text has been added.
Line 336
Done.
Line 447 cited Ketris and Yudovich (2009).
The reference has been added.
Line 492 In low ash yield coals, sometimes accessory minerals (e.g. chromite grains) could elevate V concentrations. Please, keep in mind this possibilty.
Yes, I paid attention to this aspect, but in the tested samples there were few such minerals; on the other hand, significant amounts of ash were observed.
Line 500 Did you observe any cleat/fracture infilling minerals (e.g. quartz or kaolinite) in the studied samples?
Yes, the cracks were filled with iron sulfides. This observation was added to the manuscript.
Reviewer 3 Report
In the present work, the authors investigated the chemical composition of coal ash and the content of the following critical elements: Ga, Sc, and V in coal and ash using samples of different lithotypes. The correlation analysis was applied for determining the relationships between the chemical composition of ash and the content of critical elements. The work is interesting and has certain merits. However, the text is not well arranged and the logic is not clear, in particularly English writing. There are many mistakes in the manuscript and the experimental findings are not new. So, I recommend the manuscript to be rejected. The following are the questions and some mistakes in this manuscript:
- Critical elements studied in the work are Ga, Sc, and V. However, in the abstract, Sn appeared instead of Sc.
- Before submitting, the material is not properly prepared and formatted. In line 145, 176,179, 377 and 540, redundant symbols appeared. The chemical formulas must be described precisely as well (line 543, Al2O3 appeared, where 2 and 3 need to be in the form of subscript).
- Proximate, ultimate, and petrographic results of the examined coal are presented in Table 2. However, the authors failed to provide clear explanation of the results.
- In line 279, a conclusion from a reference was cited. However, it is not indicated which literature is cited.
Author Response
The authors would like to thank for valuable comments and linguistic corrections.
Some of the specific comments need to be answered:
In the present work, the authors investigated the chemical composition of coal ash and the content of the following critical elements: Ga, Sc, and V in coal and ash using samples of different lithotypes. The correlation analysis was applied for determining the relationships between the chemical composition of ash and the content of critical elements. The work is interesting and has certain merits. However, the text is not well arranged and the logic is not clear, in particularly English writing. There are many mistakes in the manuscript and the experimental findings are not new.
The experimental findings are new due to the fact that the relationships between the individual parameters of coal and its chemical composition were examined in terms of different lithotypes. So far, similar studies have been carried out on the entire seam without detailed petrographic analysis. At the same time, it should be noted that the research concerns selected critical elements that are intensively sought after all over the world. Therefore, the proposed method of applying correlation analysis can be used for a preliminary assessment of the economic viability of extraction of selected elements.
Two subsections on proximate and ultimate analysis of coal and petrographic analysis have also been added
The English language was checked by English Editing Publishing Services (MDPI), English editing ID: english-21773.
So, I recommend the manuscript to be rejected. The following are the questions and some mistakes in this manuscript:
- Critical elements studied in the work are Ga, Sc, and V. However, in the abstract, Sn appeared instead of Sc.
Thank you for your valuable comment. The mistake has been corrected.
- Before submitting, the material is not properly prepared and formatted. In line 145, 176,179, 377 and 540, redundant symbols appeared. The chemical formulas must be described precisely as well (line 543, Al2O3 appeared, where 2 and 3 need to be in the form of subscript).
The mistakes have been corrected.
- Proximate, ultimate, and petrographic results of the examined coal are presented in Table 2. However, the authors failed to provide clear explanation of the results.
- In line 279, a conclusion from a reference was cited. However, it is not indicated which literature is cited.
The reference has been added.
The language and punctuation have been corrected and new chapters have been added.
The manuscript was checked by English Editing Publishing Services (MDPI), English editing ID: english-21773.
I hope that the changes and improvements made will satisfy the Reviewer and the manuscript will be acceptable for publication.
Round 2
Reviewer 2 Report
The author did most of suggested corrections. However, some minor points are still existing. In table 2, it is better to report one digit after comma for proximate results. Similarly, in table 3 please report one digit after comma and just leave blank results for not observed minerals rather than 0.000. Finaly, in the reference list please provide authors names and surnames for cited study with number 72. Overall, after the suggested corrections done, the MS will be ready for publication.
My personal suggestion to author for her future studies, she should try to use SEM-EDX or another imaging methods in order to have more robust results for mode of occurrences of critical elements.
This manuscript is a resubmission of an earlier submission. The following is a list of the peer review reports and author responses from that submission.
Round 1
Reviewer 1 Report
The work under consideration is titled “Ash characteristics and selected critical elements (Ga, Sc, V) in coal and ash in Polish deposits”. However a significant part of the manuscript reviews the industrial use of critical elements without any reference to the Polish coal deposits.
The study is based on the proximate, ultimate and petrographic analyses in addition to ICP-MS for a few trace elements. It is not explained, why the ICP-MS study was so limited. No XRD or other mineralogical analyses have been undertaken. As a result, there are some mistakes in description of the ash mineralogy.
The sediments sources of studied coals have not been discussed, excluding a mention of volcanic source of vanadium. Meanwhile, Ga, Sc, and V are suggested to have inorganic source.
The limited data result in many speculations in the discussion and conclusion sections.
More distinct comments may be found in the attached PDF file.
Good sides of the manuscript are a clear language and good quality of tables and figures, which, however, cannot hide a low quality of research as itself.

Author Response
All the comments included in the revision have been included.
The authors would like to thank for valuable comments and linguistic corrections.
Some of the specific comments need to be answered:
- The work under consideration is titled “Ash characteristics and selected critical elements (Ga, Sc, V) in coal and ash in Polish deposits”. However a significant part of the manuscript reviews the industrial use of critical elements without any reference to the Polish coal deposits.
There is no reference to the Polish coal deposits because the recovery of elements from ash is currently not performed in Poland. There are several projects on the subject, but their results are not satisfactory.
The study is based on the proximate, ultimate and petrographic analyses in addition to ICP-MS for a few trace elements. It is not explained, why the ICP-MS study was so limited. No XRD or other mineralogical analyses have been undertaken. As a result, there are some mistakes in description of the ash mineralogy.
A full ICP-MS analysis (53 elements) was conducted. However, the intention was to focus on a few selected ones. Originally, I wanted to describe 9 elements, but it turned out that the paper would be too detailed and too long. Therefore, I selected only three of them. It should be mentioned that the data on V and Sc in coal and ashes are relatively limited, which is why I decided to write about these elements.
XRD analysis was performed. XRD analysis showed typical minerals found in coal and ash. Due to the fact that several dozen samples were analyzed, no charts with this analysis were included.
It has been added that XDR analysis was performer. The results for lignite and bituminous coal are now included.
- The sediments sources of studied coals have not been discussed, excluding a mention of volcanic source of vanadium. Meanwhile, Ga, Sc, and V are suggested to have inorganic source.
The mineral matter in Polish coal deposits has a similar genesis to that of other coal deposits worldwide. Due to the fact that the study used coal samples from 10 deposits, the exact geological characteristics and genesis of each were not discussed.
The origin of the mineral matter in the studied deposits is now discussed, although in many cases their origin is not entirely known.
- The limited data result in many speculations in the discussion and conclusion sections.
The presented results are part of a larger project, which is why the conclusions are also based on the experience gained. The results of other analyzes have also been added.
Comments from the PDF file
- Line 84 The first basic studies were made by the Russian researchers (see references in Seredin and Dai, 2012).
New references have been added
- Line 98 How the samples were powdered. Was polution from the mill excluded?
The samples were pulverized using ball mills. All studies were supervised by the author and all procedures were followed to prevent contamination of samples.
- Line 127 What about aluminosilicates?
Naturally, aluminosilicates can also be found. Blame it on brachylogical way of writing. The sentence has been deleted.
- Line 168 Clays are decomposed in ash, aren't they?
The XRD analysis showed, among others, illite and clay minerals. The description of the XDR analysis has been added.
- Line 296 Au, Ag and PGE are commonly termed as precious.
The sentence has been rewritten, “precious” is now removed
- Line 332 clays are also aluminosilicates. Do you like to say "clay minerals and other aluminosilicates"?
The “Clay minerals” part has been removed.
- Line 335 What are reasons for these suggestions?
A literature reference has been added.
- Line 340 Ga content in coal significantly depends on its content in sediment source and/or hydrothermal influence. Without studying these matters, you can not make this conclusion.
In Polish deposits there are areas where hydrothermal influence is clear; there are also Zn-Pb deposits in close proximity to coal deposits. However, in the Bogdanka deposit, where the highest concentrations are observed, there are no hydrothermal deposits. However, based on the results of my previous studies on sulfides in the Bogdanka deposit, it can be stated that sulfide mineralization is epigenetic in relation to coal deposits.
- Line 372 in ash, not in lignite and coal
The sentence has been corrected.
- Line 380 How about sulfides? Ga is positively correlated with inorganic matter and V. Why V is associated with organic matter?
No correlation with sulfides was shown, but perhaps it is significant. Based on observations it was found that there is a correlation with sulfur, but most likely it is organic sulfur associated with organic matter. Ga correlates with V because there is a geochemical affinity between them. Based on the analysis, it can be presumed that the enrichment in Ga and V was from the same source; Ga absorbed e.g. clay minerals and V was bound with the organic matter. However, this statement would have to be confirmed, for example, by means of electron microprobe analysis and requires further research.
- Line 468 In the text above, the authors suggested that V is associated with organic matter
The sentence has been corrected.

Reviewer 2 Report
The article titled: „Ash characteristics and selected critical elements (Ga, Sc, V) in coal and ash in Polish deposits” present report of examination of coal ash of selected sample terms to critical elements. In my opinion it can be worthly to note article but after major revision according to following remarks:
Introduction:
- also information about HREY LREY and REE and REY should by explain in this part of article.
- In the sentence: „The question of critical raw materials in coal has been discussed by…” should contain listed names of the authors not only numer of references. The same next sentence.
- at the end of the introduction chapter there is a lack of information about novelty of the article.
Materials and Methods
- line 106 please give subscripts in „HNO3:HCl:H2O”
Results and discussion
- line 201 remove coma after „problematic”.
- Line 218 please give references „[34]” before dot.
- Most of information from subchapters 3.2.1, 3.2.2 3.2.3 are theoretical they should be place in introduction chapter.
- I have doubt if verification corelation of critical elements in Al2O3 is realiable. In my opinion also correlation between mineral composition of coals (clay minerals, feldspars etc.) should be verified in form of graphs. General some more detail analysis is needed
Conclusion
- This chapter should indicate further plans of examination in order to show utility of presented works.
So resume in the article author should underline novelty because I didn’t see it as well as expand analysis of correlation between critical elements and mineral composition.
Author Response
All the comments included in the revision have been included.
The authors would like to thank for valuable comments and linguistic corrections.
Some of the specific comments need to be answered:
Introduction:
- also information about HREY LREY and REE and REY should by explain in this part of article.
HREY and LREY were discussed, see from line 43 on. These elements were not discussed in detail because they are not important for the study.
In the sentence: „The question of critical raw materials in coal has been discussed by…” should contain listed names of the authors not only numer of references. The same next sentence.
The references have been rewritten according to the guidelines.
- at the end of the introduction chapter there is a lack of information about novelty of the article.
The novelty of the article lies in the analysis of elements broken down by lithotype; this is now underlined at the end of the introduction chapter.
Materials and Methods
- line 106 please give subscripts in „HNO3:HCl:H2O”
This has been corrected.
Results and discussion
- line 201 remove coma after „problematic”. –
This has been corrected.
- Line 218 please give references „[34]” before dot.
This has been corrected.
- Most of information from subchapters 3.2.1, 3.2.2 3.2.3 are theoretical they should be place in introduction chapter.
It was decided to include this information to maintain the consistency of the text. For a reader looking for information on a particular element, this is a great help as all the data are in one section.
- I have doubt if verification corelation of critical elements in Al2O3 is realiable. In my opinion also correlation between mineral composition of coals (clay minerals, feldspars etc.) should be verified in form of graphs. General some more detail analysis is needed
I agree that an in-depth analysis is necessary; however, it should be borne in mind that petrographic analysis and determining the percentage composition of mineral matter and minerals is subject to some error. This is due to the fact that clay minerals impregnate macerals, which is why they are not fully visible in the microscopic image. Therefore, we are unable to determine their quantitative share. Thus, in my opinion, the results of chemical analysis are better suited for statistical analysis. The petrographic analysis determined quartz, sulfides, clay minerals, and sulfates. On the other hand, the correlation analysis performed did not show statistically significant correlations. This issue is now discussed in the paper.
Conclusion
- This chapter should indicate further plans of examination in order to show utility of presented works.
So resume in the article author should underline novelty because I didn’t see it as well as expand analysis of correlation between critical elements and mineral composition.
The section has been rewritten.

Reviewer 3 Report
(Line 59) Barium is in fact a metallic element, and should be listed with the other metallic elements.
(Line 76) In 45 years of working with coals, I have never heard the term "professional power plant." What does it mean?
(Line 99) I suggest that the "accredited laboratory" be identified in this line, not identified later in line 103. I also suggest that all of the "applicable standards" either be identified or at least cited.
(Line 127) The author states that "most of the elements are in the form of oxides," which is not true in any case, but then she contradicts her own assertion just a few lines below, by mentioning gehlenite, anorthite, portlandite, and anhydrite.
(Line 159) The author should make it very clear that discussing "CaO content" does not mean that there is actually CaO in the ash or in the coal. This is simply a way of referring to the calcium content as if it were present as CaO, because the traditional way of reporting the composition of ashes (and rocks) is to pretend that each element is present as an oxide. In most cases they are not. CaO is well known to react readily with moisture and/or carbon dioxide in the environment, so would not exist in a deposit that has been in the Earth's crust for millions of years.
(Line 175) Continuing from the previous comment, there is positively no Na2O or K2O in any ash. Both of these compounds react violently with water.
(Lines 179-187) I suggest that the author should recall that "correlation does not imply causation." She has obviously worked hard to analyze the numerical data. However, it seems that some (granted, not all) of the correlations reported are simply adventitious correlations from the "number crunching." The statement about "an increase in CaO in ash increases the volatile content" is particularly hard to accept. Just because there is an apparent correlation between these two variables, that does not mean that somehow CaO causes, or is responsible for, formation of volatile matter.
(Lines 221-222). I would call the author's attention to a typing error: there is no such compound as TiO3. Surely she meant TiO2.
(Line 296) Strictly speaking, gallium is not a "precious metal." In common usage, the term precious metal refers to the platinum group metals plus gold and silver.
(Line 306) The author's use of the term Clarke values is confusing. As normally used, the Clarke value for an element refers to its average concentration in the Earth's crust. Therefore I don't see how an element could have a Clarke value for coal, another one for ash, etc. Perhaps the author is meaning to talk about enrichment factors, but those are dimensionless numbers (concentration in the sample divided by the Clarke for that element). And, if the author has not actually calculated enrichment factors, it would be an extremely useful exercise to undertake, and would strengthen the manuscript greatly.
Author Response
All the comments included in the revision have been included.
The authors would like to thank for valuable comments and linguistic corrections.
Some of the specific comments need to be answered:
(Line 59) Barium is in fact a metallic element, and should be listed with the other metallic elements.
This has been changed and barium is now listed with the other metallic elements
(Line 76) In 45 years of working with coals, I have never heard the term "professional power plant." What does it mean?
The sentence has been changed. It applied to large, industrial power plants.
(Line 99) I suggest that the "accredited laboratory" be identified in this line, not identified later in line 103. I also suggest that all of the "applicable standards" either be identified or at least cited.
The laboratory name and standards, according to which individual parameters were determined, have been added.
(Line 127) The author states that "most of the elements are in the form of oxides," which is not true in any case, but then she contradicts her own assertion just a few lines below, by mentioning gehlenite, anorthite, portlandite, and anhydrite.
The sentence has been removed.
(Line 159) The author should make it very clear that discussing "CaO content" does not mean that there is actually CaO in the ash or in the coal. This is simply a way of referring to the calcium content as if it were present as CaO, because the traditional way of reporting the composition of ashes (and rocks) is to pretend that each element is present as an oxide. In most cases they are not. CaO is well known to react readily with moisture and/or carbon dioxide in the environment, so would not exist in a deposit that has been in the Earth's crust for millions of years.
(Line 175) Continuing from the previous comment, there is positively no Na2O or K2O in any ash. Both of these compounds react violently with water.
Thank you for this valuable comment. It was clear to me that the oxide composition presented did not directly reflect the mineral composition, however, some readers could see it that way. Therefore, it has been stressed out that the oxide composition is only a chemical composition without a direct impact on minerals.
(Lines 179-187) I suggest that the author should recall that "correlation does not imply causation." She has obviously worked hard to analyze the numerical data. However, it seems that some (granted, not all) of the correlations reported are simply adventitious correlations from the "number crunching." The statement about "an increase in CaO in ash increases the volatile content" is particularly hard to accept. Just because there is an apparent correlation between these two variables, that does not mean that somehow CaO causes, or is responsible for, formation of volatile matter.
The remark that the correlation does not imply causation was added. However, in my opinion, increasing the carbonate content in the coal, which results in increased CaO content in the ash, increases the volatile matter content. The volatile matter includes decomposing calcite. Although the temperature of its total decomposition is higher than 850oC, the breakdown of CaCO3 into CaO and CO2 is already visible at a temperature of about 500oC.
(Lines 221-222). I would call the author's attention to a typing error: there is no such compound as TiO3. Surely she meant TiO2.
Yes, it is a typo. Thank you for pointing it out. I meant TiO2.
(Line 296) Strictly speaking, gallium is not a "precious metal." In common usage, the term precious metal refers to the platinum group metals plus gold and silver.
The term precious has been removed.
(Line 306) The author's use of the term Clarke values is confusing. As normally used, the Clarke value for an element refers to its average concentration in the Earth's crust. Therefore I don't see how an element could have a Clarke value for coal, another one for ash, etc. Perhaps the author is meaning to talk about enrichment factors, but those are dimensionless numbers (concentration in the sample divided by the Clarke for that element). And, if the author has not actually calculated enrichment factors, it would be an extremely useful exercise to undertake, and would strengthen the manuscript greatly.
The Clarke value for coal and ash was determined by Ketris, M. P.; Yudovich, Y. E. Estimations of Clarkes for Carbonaceous biolithes: World averages for trace element contents in black shales and coals. International Journal of Coal Geology 2009, 78, 135–148, 589 doi:10.1016/j.coal.2009.01.002.. At this moment, this work is cited almost 900 times. In my scientific work, I've repeatedly stumbled upon the terms Clarke value in the earth's crust, Clarke value in sedimentary rocks, etc. Therefore, there should be no ambiguity here.
As for enrichment factors, they are included in the Table and are briefly addressed in the paper.

Reviewer 4 Report
The related discussions were done and organization of MS is suitable for publication; however, some points need to be re-organize by author. Firstly, the results of standard coal quality and petrographical compositions should be added as a separate subchapter in results and discussion chapter. Furthermore, if it is possible, author should provide some snaps of macerals and mineral matter from coal petrography analysis. The latter one could be useful for mineralogy and mode of occurrence of selected elements. Secondly, the author should provide a table which is reporting the identified minerals in the studied coal and ash samples. Finally, the author should summarize and provide less literature data about Ga, V and Sc data in the related subchapters. I added some suggestions and corrections in the attached revised MS. Overall, I would like to re-consider the MS, after suggested corrections done.

Reviewer 5 Report
Reviewer`s remarks concerning the paper:
“Ash characteristics and selected critical elements (Ga, Sc, V) in coal and ash in Polish deposits”
The subject matter brought up in the article is interesting. The article contains interesting results but reviewer has some comments.
Lines 122-125 –the procedure and software used to determine the content of mineral phases (quantitative XRD analyse) should be given; for which samples the XRD analysis was performed - for coal, ash, or both coal and ash.
Lines 133-142 - the results of the quantitative XRD analysis should be presented in the table.
Lines 207-208 - were the results compared for all ash samples or only for lignite ash?
Lines 212-221 - Appendix 1 - in addition to the correlation coefficient r, the significance level p should be shown. Only significant correlations should be analysed. Correlation coefficient and significance level values should be given to two decimal points.
On the basis of these correlations, it should be possible to explain the relationships between the parameters characterizing the samples. The relationships shown in figures 1-3 and presented in the text should be explained.
Table 3 there is no unit at Al2O3.
Lines 222-223 and 235 -236 – What is the influence of the phase composition of the mineral substance in coal on the presence of these oxides and the properties of ash? What does mean moderate effect on softening (tA), melting (tB) and fluid (tC) points.
Lines 249-301– The indexes: Base/Acid (B/A) ratio, Slagging Index (Rs), Silica Value (SV), Fouling Index (Rf), Alkalinity (AK) for all ash samples should be calculated and presented in the table. On the basis of the values of these indexes, the ashes should be classified.
Line 262 - the total sulphur content in coal, dry basis symbol is Std or Stdb (table 2)?
In table 4, apart from the average contents of Ga, Sc, V and the values of enrichment factors in coal and ash the standard deviation should be given.
There are also some discrepancies between values in the text and in the table 4:
line 345 average Ga content in bituminous coal
line 351 average Ga content in bituminous coal ashes
line 467 average Sc content in coal and in lignite
line 470 average Sc content in bituminous coal ashes
Lines 376-377 - How should be interpreted moderate negative correlations between the Ga content in ash and the total porosity of coal and volatile matter content?
Lines 425-426 there is the sentence: “The average vanadium content for the tested samples was 458.2 ppm, with a high standard deviation of 429.1 ppm.” The values presented in the text are not included in table 4.
For the comfort of potential readers, all analysed significant correlations should be shown on collective figures, e.g. small graphs in the form of a plate, r and p values should be included, in relevant graphs.
It would be better, to include in the text samples numbers (e.g. 1L, 2B) next to deposit name
Reference list - There is no citation in the text for publication number 3.
Conclusions should be corrected after taking into account all research results.
Round 2
Reviewer 1 Report
I am not satisfied with the author’s responses to my major comments. These are unsubscribing rather than adequate answers. Who may be interested in knowingly limited information? Why publishing dozens of analyzes is a problem? Significant additional data can overturn all early conclusions. If the study is not completed, then why publish it.
“The mineral matter in Polish coal deposits has a similar genesis to that of other coal deposits”. “Other coal deposits worldwide” are highly different from each other.
Perhaps, the author’s aim is just to show that ashes from the Polish coals may be utilized industrially. However, even the industry needs much more geochemical and mineralogical information to get interested.
I have made my responses to some formal questions more negative here than in my previous review, since the author showed in the current draft that some significant data (XRD, multi-element ICP-MS) on the matter have been obtained, but not presented and adequately discussed.
Reviewer 2 Report
After correction article can be publish after accptance of Editor.